# Decision Theoretic Foundations for Conformal Prediction: Optimal Uncertainty Quantification for Risk-Averse Agents

Shayan Kiyani [1]  George Pappas [1]  Aaron Roth [1]  Hamed Hassani [1]

## Abstract

A fundamental question in data-driven decision making is how to quantify the uncertainty of predictions to inform risk-sensitive downstream actions, as often required in domains such as medicine. We develop a decision-theoretic foundation linking prediction sets to risk-averse decision-making, addressing three questions: (1) What is the correct notion of uncertainty quantification for risk-averse decision makers? We prove that prediction sets are optimal for decision makers who wish to optimize their value at risk. (2) What is the optimal policy that a risk averse decision maker should use to map prediction sets to actions? We show that a simple max-min decision policy is optimal for risk-averse decision makers. Finally, (3) How can we derive prediction sets that are optimal for such decision makers? We provide an exact characterization in the population regime and a distribution free finite-sample construction. These insights leads to *Risk-Averse Calibration (RAC)*, a principled algorithm that is both *practical*—exploiting black-box predictions to enhance downstream utility—and *safe*—adhering to user-defined risk thresholds. We experimentally demonstrate RAC's advantages in medical diagnosis and recommendation systems, showing that it substantially improves the trade-off between safety and utility, delivering higher utility than existing methods while avoiding critical errors.

## 1. Introduction

Predictions are frequently used to inform *actions*. For example, in clinical medicine, patient data are used to predict diagnoses and outcomes when choosing treatments. In high-stakes cases—where an incorrect treatment decision could lead to serious complications or death—it is crucial not to rely solely on a model's predictions. Instead, decisions must account for the uncertainty in these predictions, opting for more conservative interventions when that uncertainty makes the potential outcomes (e.g., complications, side effects) highly variable. Connecting uncertain predictions to actionable, principled decisions is a significant challenge in safety-critical domains, including medical diagnosis, finance, robotics, and control, and requires balancing safety with utility. One extreme is to avoid any action entirely—sacrificing prediction's practical value for absolute safety—while the other is to aggressively exploit predictions to maximize expected utility, accepting significant downside risk at the cost of realizing poor outcomes with substantial probability. Balancing this trade-off calls for an optimal approach to risk-sensitive decision making. To this end, we focus on the following question:

*What is the optimal interface between prediction and action that allows for navigating the trade-off between safety and utility in high stakes applications?*

The optimal design of an action policy crucially depends on how uncertainty is quantified. Among various methods, a widely adopted approach—spurred by advances in conformal prediction—is to produce *prediction sets* rather than point estimates. But what exactly are prediction sets good for? Which decision-making processes make them the right language for communicating uncertainty? And, given such a process, what is the optimal rule for transforming prediction sets into actions? To address these questions, we first introduce our setting and notation. We consider a feature space $\mathcal{X}$ and a label set $\mathcal{Y}$, endowed with the distribution $(x, y) \sim \mathcal{D}$. A downstream decision maker has an action set $\mathcal{A}$ and a utility function $u : \mathcal{A} \times \mathcal{Y} \to \mathbb{R}$ that maps actions $a$ and realized labels $y$ to utilities $u(a, y)$, which the decision maker seeks to maximize. Upon observing $x \in \mathcal{X}$, the decision maker must take an action $a \in \mathcal{A}$ without observing the true label $y$, relying instead on predictions about $y$. Within this framework, we aim to answer the above questions.

In seeking answers, it is instructive to reflect on what we can say about *calibrated forecasts*, an alternative way of quantifying uncertainty with well-established decision-theoretic

[1]Department of Electrical and Systems Engineering, University of Pennsylvania, Philadelphia, USA. Correspondence to: Shayan Kiyani <shayank@seas.upenn.edu>.

*Proceedings of the 42nd International Conference on Machine Learning*, Vancouver, Canada. PMLR 267, 2025. Copyright 2025 by the author(s).

## Risk Averse Calibration (RAC)

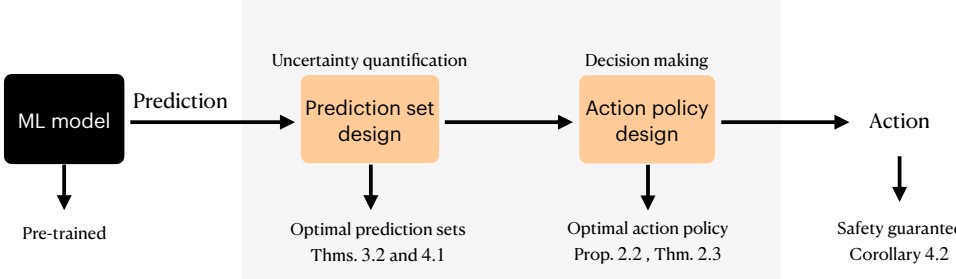

*Figure 1.* RAC pipeline, an interface between prediction and action for high-stakes applications.

foundations—that has its own limitations. Suppose we are in a multiclass classification setting, and we represent labels $y \in \mathcal{Y}$ using one-hot vectors in the $k$-dimensional probability simplex. A forecasting rule $f : \mathcal{X} \to [0, 1]^k$ is *calibrated* if, for every prediction $\hat{p}$, we have $\mathbb{E}[y \mid f(x) = \hat{p}] = \hat{p}$, meaning it is unbiased given the forecast. Then a simple consequence of calibration (Foster and Vohra, 1997; Zhao et al., 2021; Noarov et al., 2023) is that for any *expectation maximizing* decision maker, choosing the action that would maximize expected utility as if the forecast was correct is the optimal policy amongst all policies mapping forecasts to actions. Formally, if $f$ is calibrated, then applying $BR_u(f(x)) = \arg\max_{a \in \mathcal{A}} \mathbb{E}_{y \sim f(x)}[u(a, y)]$ achieves higher expected utility than any other policy mapping forecasts to actions. In this sense, calibration is the right language for communicating uncertainty to *expectation maximizing—i.e. risk neutral—agents*, and the right rule for such agents to ingest calibrated forecasts is to act as if they are correct specifications of the label distribution.

In contrast, we seek the right interface between predictions and actions for *risk-averse agents*. Let $a(\cdot) : \mathcal{X} \to \mathcal{A}$ be an action policy. We call $\nu(\cdot) : \mathcal{X} \to \mathbb{R}$ a *utility certificate* if it satisfies the following *safety guarantee*:

$$\Pr[u(a(X), Y) \geq \nu(X)] \geq 1 - \alpha. \tag{1}$$

In words, with probability at least $1 - \alpha$, the utility of an agent following the policy $a(x)$ is guaranteed to be at least $\nu(x)$. Naturally, we aim to maximize the average value of the utility certificate $\nu$ subject to satisfying the requirement in (1) – i.e., as the risk-averse agent, we seek to maximize the average quantile of their utility, commonly referred to as the *value at risk* in the financial risk literature (see Section 2 for details on the problem formulation). This objective yields the optimal balance between safety and utility, achieved by finding the pair $(a, \nu)$ that satisfies the safety constraint while maximizing the average utility certificate.

In practice, however, the true probability distribution that connects the actions to their utility values is unknown. Instead, the decision maker must rely on (uncertain) predic-

tions to best balance the trade-off between safety and utility. The core challenge in this regard is to develop the right notion of uncertainty quantification for the predictions and optimal action policies based on such uncertainty measures.

We show that prediction sets are the right medium for communicating uncertainty to risk-averse decision makers who seek high-probability guarantees on their realized utility, i.e., the quantiles of their utility distribution as formulated in (1). Specifically, we prove that optimizing action policies to maximize utility while satisfying (1) is fundamentally equivalent to designing *prediction sets* optimally, followed by a simple max-min decision rule. This establishes prediction sets as a sufficient statistic for safe action policies, encapsulating all necessary information for risk-averse decision making. We then derive an explicit formulation for the optimal prediction sets, which serves as the foundation for a finite-sample algorithm providing distribution-free safety guarantees. Put together, these results characterize the optimal interface between predictions and actions for risk-averse decision making as depicted in Figure 1. In more detail:

1. **Max-min decision rule.** When given prediction sets $C(x)$ with only a *marginal* coverage guarantee, risk-averse decision makers should choose their action by maximizing worst-case utility over all labels $y \in C(x)$. We prove this *max-min* policy is minimax optimal over all data distributions satisfying the marginal coverage guarantee (Proposition 2.2).

2. **Prediction-set equivalence.** The optimal pair of action policy and utility certificate can be obtained by applying the max-min decision rule to a suitably designed prediction set with marginal coverage (Theorem 2.3). This establishes that prediction sets are a sufficient statistic for safe decision making.

3. **Optimal design of prediction sets.** We formulate *Risk Averse Conformal Prediction Optimization* (Section 2.2) to find prediction sets that maximize the target utility quantile under the max-min policy. Using duality theory, we derive an explicit, one-dimensional char-

acterization of the optimal sets (Theorem 3.2), which underpins our finite-sample construction.

4. **Finite-sample algorithm.** We propose *Risk-Averse Calibration (RAC)* (Section 4), which can exploit any black-box predictive model to derive action policies and utility certificates while providing a *distribution-free* safety guarantee (1). This guarantee holds for any given utility function.

5. **Experiments.** In Section 5, we compare RAC with several conformal-prediction methods (Cortes-Gomez et al., 2024; Romano et al., 2020; Sadinle et al., 2019) and best response baselines. Across multiple tasks, such as medical diagnosis, RAC achieves a superior trade-off between safety and utility, delivering higher utility at each user-specified risk threshold.

## 1.1. Related Work

Conformal prediction (CP), introduced by Vovk et al. (2005), provides a flexible framework for constructing prediction sets with finite-sample guarantees (Lei et al., 2018; Shafer and Vovk, 2008). Recent research has explored adapting CP to various decision-making problems. Here, we briefly discuss the most relevant works, and provide a thorough discussion in the Appendix A.

**Risk Control.** A growing line of research extends CP beyond coverage constraints to control more general risk measures (Lindemann et al., 2023; Angelopoulos et al., 2022; 2021; Cortes-Gomez et al., 2024; Lekeufack et al., 2024). Angelopoulos et al. (2022) propose conformal risk control for risk measures over prediction sets, and Cortes-Gomez et al. (2024) extend this by constructing sets that satisfy coverage while achieving low risk. However, they do not explicitly discuss which actions their sets should inform or how to design these sets to best serve the decision maker. Lindemann et al. (2023) apply conformal prediction to safe planning, and Lekeufack et al. (2024) focus on decisions parameterized by a single scalar, calibrated to control risk. However, they restrict their action policy to a *predefined* low-dimensional family, leaving open the question of how to *jointly* optimize over policy design and uncertainty quantification for risk-averse utility.

In this paper, we fill this gap by addressing three core questions for a risk-averse decision maker: (1) *What is the correct notion of uncertainty quantification?* We prove that prediction sets are optimal for high-stakes decisions. (2) *How can we design these optimal sets?* We provide an exact population-level characterization and a distribution-free, finite-sample construction. (3) *What is the optimal policy given these sets?* We show that a simple max–min rule is optimal for risk-averse utility. In Section 5, we implement the most recent approach in this direction, Cortes-Gomez et al. (2024) and demonstrate that our framework yields

significantly more effective action policies.

**Risk Aversion in Economics.** Decision-making under risk aversion is fundamental in economics, beginning with Bernoulli's expected utility theory (Bernoulli, 1954) and formalized by Von Neumann and Morgenstern's axiomatic model (von Neumann and Morgenstern, 1944). Pratt (Pratt, 1964) and Arrow (Arrow, 1965) introduced precise measures of risk aversion (Arrow–Pratt coefficients), while Hadar and Russell (Hadar and Russell, 1969) and Hanoch and Levy (Hanoch and Levy, 1969) developed stochastic dominance criteria. Rothschild and Stiglitz (Rothschild and Stiglitz, 1970) further refined risk comparison through mean-preserving spreads. Recent extensions have addressed robust criteria such as maximin and minimax-regret under ambiguity (Manski, 2000; 2004; Manski and Tetenov, 2007; Manski, 2011) (see also the recent survey (Royset, 2024)). Unlike these classical frameworks, our approach emphasizes data-driven learning and distribution-free uncertainty quantification, providing risk-averse guarantees applicable to any black-box pretrained model.

**Domain-Specific CP Methodologies.** Decision making with CP has also been explored in specific domains such as robust optimization (Patel et al., 2024b; Johnstone and Cox, 2021; Yeh et al., 2024), medical tasks (Banerji et al., 2023; Vazquez and Facelli, 2022), power and energy systems (Renkema et al., 2024), formal verification (Lindemann et al., 2024), and chance-constrained optimization (Zhao et al., 2024). While our framework could potentially be extended to these settings, each may involve additional domain-specific challenges beyond the scope of this work. Additionally, recent works also explored the application of CP sets in decision making in the context of counterfactual inference (Lei and Candès, 2021; Yin et al., 2024; Jin et al., 2023). We, however, focus on risk averse decision making using prediction sets. In particular, we show that prediction sets are a sufficient statistic for risk averse agents that aim to optimize their value at risk.

## 2. The Preliminaries of Risk-Averse Decision Making

In this section, we will formalize the central objective of a risk averse decision maker. Recall that in our stetting, upon observing $x \in \mathcal{X}$, the decision maker will have to take an action $a \in \mathcal{A}$. The decision maker does not observe the true label $y$, but its utility will depend on both the action $a$ and label $y$, which is captured by a given utility function $u$.

We focus on *risk-averse* decision making, where the goal is to choose actions that ensure a sufficiently high utility *with high probability* over the randomness of the label. That is, risk aversion prioritizes minimizing the likelihood of low-utility outcomes, even at the cost of overlooking higher but uncertain utilities. Formally, given a risk tolerance thresh-

old $\alpha$, a decision maker facing $x \in \mathcal{X}$ assigns each action $a \in \mathcal{A}$ a value: $\nu_\alpha(a; x) := \text{quantile}_\alpha[u(a, Y) \mid X = x]$, where $Y \sim p(y \mid x)$. This standard risk measure, known in financial risk literature as *Value at Risk (VaR)* (Duffie and Pan, 1997), represents the largest value such that, if action $a$ is taken, the utility is at least $\nu_\alpha(a; x)$ with probability $1 - \alpha$. Thus, the risk-averse decision maker selects the action maximizing $\nu_\alpha(a; x)$, ensuring the highest guaranteed utility.

$$\nu_\alpha(x) = \max_{a \in \mathcal{A}} \nu_\alpha(a; x) \qquad (2)$$
$$:= \max_{a \in \mathcal{A}} \text{quantile}_\alpha \left[ u(a, Y) \mid X = x \right], \quad \forall x \in \mathcal{X}.$$

The above *risk-averse* utility should be contrasted with the best *expected* utility $\max_a \mathbb{E}[u(a, Y) | X = x]$. The latter leads to actions that maximize the average utility whereas the former aims to maximize the worst-case utility that can happen with probability $1 - \alpha$. Hence the former will be more risk averse at the cost of becoming more conservative. It is important to mention that the economic literature extensively explores various other notions of risk aversion, such as Conditional Value-at-Risk (CVaR) (see e.g. (Royset, 2024)). However, here we only focus on the aforementioned risk measure, and the exploration of these alternative risk notions remains beyond the scope of this work.

**Marginal Version.** The quantity in (2) is a *point-wise* or *conditional* quantity; i.e. to find the best action according to (2) the decision maker requires access to the conditional distribution $p(y|x)$. In practice, such distributions are unknown, and guarantees of the form (2) are often intractable when only a finite sample of the distribution is available. An analogous situation arises in conformal prediction (CP), where obtaining fully-conditional coverage guarantees is known to be impossible from a finite sample of data. Consequently, conformal prediction focuses on relaxed, i.e. marginal (or "group conditional", which still marginalize over part of the distribution (Bastani et al., 2022; Jung et al., 2023)) coverage guarantees which are statistically tractable.

By analogy, we will now introduce the marginal version of (2). First we rewrite the objective. For a given $x \in \mathcal{X}$, the value $v_\alpha(x)$ in (2) can be equivalently written as follows

Maximize $\nu$
$\quad a \in \mathcal{A}, \nu \in \mathbb{R}$

subject to $\quad \Pr[u(a, Y) \geq \nu \mid X = x] \geq 1 - \alpha.$

Let us examine the constraint in the above optimization more carefully. We are looking for action-value pairs $(a, \nu)$ such that we are guaranteed with probability at least $1 - \alpha$ that, when taking action $a$, the resulting utility is at least $\nu$. Of course, to maximize utility, we should maximize over the choice of the action $a$ and the value $v$ which results in the above optimization. Now, the risk-averse constraint in the above optimization has the following marginal counterpart:

$$\Pr[u(a(X), Y) \geq \nu(X)] \geq 1 - \alpha, \qquad (3)$$

where the function $a(\cdot) : \mathcal{X} \to \mathcal{A}$ is a decision-policy that[1] maps features to actions such that it guarantees average utility according to the function $\nu(\cdot) : \mathcal{X} \to \mathbb{R}$ with probability at least $1 - \alpha$, marginalized over $X$. Now, rather than optimizing over a single value for $a$ and $\nu$ for each $x$ separately, we jointly optimize over policies $a(\cdot)$ and value functions $\nu(\cdot)$[2] which map $\mathcal{X}$ to actions and values respectively. This results in the following marginal version of the decision maker's optimization problem:

---

**Risk Averse Decision Policy Optimization (RA-DPO):**

$$\underset{a(\cdot), \nu(\cdot)}{\text{maximize}} \quad \mathbb{E}_X \left[ \nu(X) \right],$$

$$\text{subject to} \quad \Pr \left[ u(a(X), Y) \geq \nu(X) \right] \geq 1 - \alpha.$$

---

**Remark 2.1.** *While our primary focus is on the marginal formulation of risk-averse optimization, one can also consider the more advanced setting of group-conditional validity (Jung et al., 2023; Gibbs et al., 2023). Specifically, for arbitrary groups $g_1, \ldots, g_m \subseteq \mathcal{X}$, the marginal constraint in RA-DPO generalizes to:* $\Pr \left[ u(a(X), Y) \geq \nu(X) \mid X \in g_i \right] \geq 1 - \alpha, \ \forall i \in [m]$. *Such constraints enable finer control over risk across subpopulations—critical in applications requiring group-specific guarantees. We leave the exploration of this objective to future works and believe our findings provide a principled first step toward that direction.*

### 2.1. A Prediction Set Perspective

Recall that in our setting the (feature, label) pair is generated according to a distribution. The decision maker only observes the feature $x$ based on which it will choose its action $a$. However, the realized utility will depend on both the action $a$ and the label $y$. The decision maker does not observe the label, but we assume that it has access to a predictor that provides predictions about the label $y$ given the input feature $x$. More specifically, we assume that the predictor will provide *prediction sets* of the form $C(x) \subseteq \mathcal{Y}$, $x \in \mathcal{X}$, that are guaranteed to contain the true label with high probability. We assume that the prediction sets satisfy the *marginal* coverage guarantee, i.e.,

$$\Pr_{(X,Y) \sim \mathcal{P}}[Y \in C(X)] \geq 1 - \alpha. \qquad (4)$$

Given this framework, two immediate questions arise: (i) Assuming the only information that the decision maker has about the true label is through the prediction sets, how should it choose its actions to maximize (risk-averse) utility? (ii) How should the prediction sets be designed to not

---

[1] In this paper, we focus on deterministic action policies.

[2] Here, note that since $\nu(x)$ is a utility function, its value can not be larger than the maximum achievable utility; i.e. $\nu(x) \leq u_{\max} := \max_a \max_y u(a, y)$ for all $x \in \mathcal{X}$.

only be marginally valid according to (4) but also maximize the utility achieved by the decision maker?

We will proceed with answering question (i) now, and will provide an answer to question (ii) in the subsequent sections. Assuming that the decision maker can only take actions based on the prediction sets – i.e. it has no other information about the label distributions - then its optimal decision rule takes a simple and natural form. It will have to play the action $a$ that maximizes their utility $u(a, y)$ in the worst case over labels $y \in C(x)$. We denote this optimal risk-averse (RA) decision rule by $a_{\text{RA}} : 2^{\mathcal{Y}} \to \mathcal{A}$, and the corresponding utility certificate by $\nu_{\text{RA}} : 2^{\mathcal{Y}} \to \mathbb{R}$:

$$a_{\text{RA}}\big(C(x)\big) = \arg\max_{a \in \mathcal{A}} \min_{y \in C(x)} u(a, y), \qquad (5)$$

$$\nu_{\text{RA}}\big(C(x)\big) = \max_{a \in \mathcal{A}} \min_{y \in C(x)} u(a, y). \qquad (6)$$

We will show that this decision rule is minimax optimal over the set of all distributions that are consistent with the marginal guarantee (4). Assume that the decision maker is given access to a set function $C : \mathcal{X} \to \{2^{\mathcal{Y}}\}$. Let us also define $\Omega$ as the set of all the data distributions that are consistent with the marginal guarantee; i.e. the set of all distributions $\mathcal{P}$ over $(\mathcal{X}, \mathcal{Y})$ such that, $\Pr_{(X,Y) \sim \mathcal{P}}[Y \in C(X)] \geq 1 - \alpha$. Let $\pi(\cdot) : 2^{\mathcal{Y}} \to \mathcal{A}$ be a policy that takes as input the prediction set $C(x)$ and outputs an action. Aligned with RA-DPO, the value of policy $\pi$ with respect to a joint distribution $p(x, y)$ can then be defined as:

$$\nu^*(\pi, p) = \operatorname*{maximize}_{\nu(\cdot)} \quad \mathbb{E}_{X \sim p(x)}\big[\nu(X)\big],$$

subject to $\displaystyle \Pr_{X,Y \sim p(x,y)}\big[u(\pi(C(X)), Y) \geq \nu(X)\big] \geq 1 - \alpha.$

We are now interested in the policy that is minimax optimal meaning that it can perform well with respect to the worst case distribution in $\Omega$. That is to say we want to find the policy $\pi^*$ that is the answer to,

$$\operatorname*{Maximize}_{\pi} \operatorname*{Minimize}_{p \in \Omega} \quad \nu^*(\pi, p). \qquad (7)$$

**Proposition 2.2.** *Assume $\alpha < 0.5$ and let $\pi^*(x)$ be the optimal solution to (7). Then we have,*

$$\pi^*(x) = \arg\max_{a \in \mathcal{A}} \min_{y \in C(x)} u(a, y). \qquad (8)$$

To summarize, Proposition 2.2 states when the risk averse decision maker decides based on a prediction set $C$, that contains the actual label with high probability, there is a simple, yet minimax optimal policy, $a_{\text{RA}}\big(C(x)\big)$ that guarantees the minimum utility of $\nu_{\text{RA}}\big(C(x)\big)$ with high probability. We now focus on how to design prediction sets that would be the most useful for the decision maker among all the prediction sets that provide valid marginal guarantee.

## 2.2. An Equivalent Formulation via Prediction Sets

In the previous section we argued that, when deciding based on prediction sets, the (minimax) optimal policy $a_{\text{RA}}$ and its associated value $\nu_{\text{RA}}$ are given in (5). Hence, assuming that the decision maker is playing $a_{\text{RA}}$, the prediction sets $C(x)$ should be designed to maximize the resulting utility of the decision maker while ensuring marginal coverage; I.e., the following optimization:

---

**Risk Averse Conformal Prediction Optimization (RA-CPO):**

$$\operatorname*{Maximize}_{C(.)} \mathbb{E}_X\left[\nu_{\text{RA}}(C(X))\right] := \mathbb{E}_X\left[\max_{a \in \mathcal{A}} \min_{y \in C(X)} u(a, y)\right]$$

subject to $\quad \Pr[Y \in C(X)] \geq 1 - \alpha.$

---

One might expect that the result of RA-CPO, i.e. optimizing the utility using prediction sets, would lead to a lower utility compared to the original optimization RA-DPO. This is because: (i) The policy given in (5) is a specific policy designed to be valid even for the worst-case distribution for which the prediction sets are marginally valid (see Proposition 2.2). Hence, this policy could be overly conservative; (ii) In RA-DPO the optimal action and value functions are obtained assuming full information about the data distribution, whereas in RA-CPO we require that information must be filtered through a (properly designed) prediction set representation. One might expect a-priori that passing from the actual distribution to a lossy prediction set representation would discard information that is critical to finding the optimal policy. However, the following theorem shows, perhaps surprisingly, that this is not the case; *the optimal action policy for any distribution can be represented as a max-min rule over a prediction set.*

**Theorem 2.3.** *RA-DPO and RA-CPO are equivalent. In other words, from any optimal solution of RA-DPO, denoted by $(a^*(x), \nu^*(x))$, we can construct an optimal solution $C^*(x)$ to RA-CPO with the same utility, i.e., $\mathbb{E}_X\left[\nu_{\text{RA}}\big(C^*(X)\big)\right] = \mathbb{E}_X\left[\nu^*(X)\right]$. Also, from any optimal solution of RA-CPO we can construct an optimal solution for RA-DPO with the same utility.*

**Implications.** Prediction sets are a fundamental object in risk averse decision making. In particular, the optimal strategy of a risk averse decision maker can be formulated as playing a max min strategy over a well-designed prediction set. To fully characterize such optimal policies, the first step is to derive the optimal solution to RA-CPO.

## 3. The Optimal Prediction Sets

We characterize the optimal solution (i.e., prediction sets) for RA-CPO given in (2.2) in terms of the conditional distribution $p(y \mid x)$. We begin by introducing the fundamental

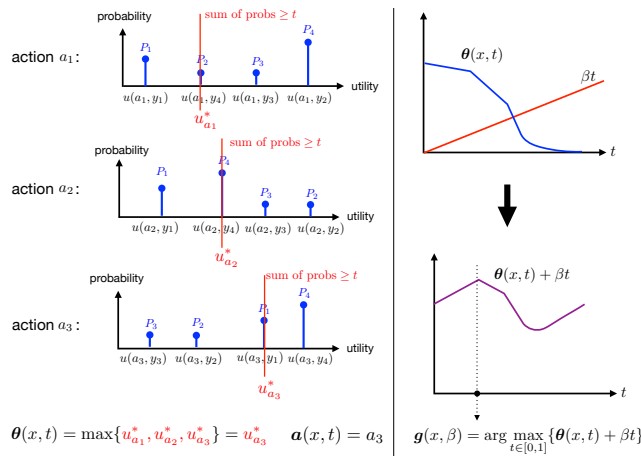

$$\boldsymbol{\theta}(x,t) = \max\{u_{a_1}^*, u_{a_2}^*, u_{a_3}^*\} = u_{a_3}^* \qquad \boldsymbol{a}(x,t) = a_3 \qquad \boldsymbol{g}(x,\beta) = \arg\max_{t \in [0,1]} \{\boldsymbol{\theta}(x,t) + \beta t\}$$

*Figure 2. Left*: Illustration of how the functions $\boldsymbol{\theta}$ and $\boldsymbol{a}$ are computed for a given $x \in \mathcal{X}$ and $t \in [0,1]$. Here we have three actions $\mathcal{A} = \{a_1, a_2, a_3\}$ and four labels $\mathcal{Y} = \{y_1, y_2, y_3, y_4\}$. We also let $P_i := p(y_i|x)$. For each of the actions $a_j$, the value $u_{a_j}^*$ is the $(1-t)$-quantile of $u(a_j, Y)$. The value $\boldsymbol{\theta}(x,t)$ corresponds to the maximum of these quantiles among the actions, and $\boldsymbol{a}(x,t)$ corresponds to the maximizing action. *Right*: Illustration of how the function $\boldsymbol{g}(x, \beta)$ is obtained from $\boldsymbol{\theta}(x,t)$ for a given $x$.

notions that relate optimal utility to coverage. We define the functions $\boldsymbol{\theta} : \mathcal{X} \times [0,1] \to \mathbb{R}$ and $\boldsymbol{a} : \mathcal{X} \times [0,1] \to \mathcal{A}$ as,

$$\boldsymbol{\theta}(x,t) = \max_{a \in \mathcal{A}} \mathrm{quantile}_{1-t}\big[u(a,Y) \mid X = x\big], \qquad (9)$$

$$\boldsymbol{a}(x,t) = \arg\max_{a \in \mathcal{A}} \mathrm{quantile}_{1-t}\big[u(a,Y) \mid X = x\big]. \qquad (10)$$

In words, given a feature $x \in \mathcal{X}$ and a probability coverage value $t \in [0,1]$, $\boldsymbol{\theta}(x,t)$ is computed as follows (see also Figure 2): For each action $a$, we first find the $(1-t)$-quantile of the random variable $u(a,Y)$ with $Y$ being distributed according to $p(y|x)$. This quantile value is the largest utility achievable with probability at least $t$ when we take action $a$. By maximizing such $(1-t)$-quantiles over the choice of the action $a$ we obtain $\boldsymbol{\theta}(x,t)$. In words, for $x \in \mathcal{X}$, the value $\boldsymbol{\theta}(x,t)$ represents the optimal (risk-averse) utility achievable under a conditional coverage assignment $t$, and the maximizing action is denoted by $\boldsymbol{a}(x,t)$.

Let us now explain how the function $\boldsymbol{\theta}(x,t)$ plays a role in finding an optimal solution for RA-CPO. Fix an instance $x$, assume that we would like to assign conditional coverage probability $t$ to $x$. For the specific instance $x$, we would like to construct a prediction set $C(x)$ that with coverage at least $t$, i.e. $\Pr(Y \in C(x) \mid X = x) \geq t$, where the probability is over the conditional distribution $p(y|x)$. We ask: How should $C(x)$ be designed to maximize the objective of RA-CPO? The following proposition provides the answer.

**Proposition 3.1.** *Fix an instance $x \in \mathcal{X}$ and a coverage value $t \in [0,1]$. Then, among all the sets $C \subseteq \mathcal{Y}$ that*

*have coverage at least $t$, i.e. $\Pr(Y \in C(x)|X = x) \geq t$, the following set has the largest risk-averse utility value $\nu_{\mathrm{RA}}(C) = \max_{a \in \mathcal{A}} \min_{y \in C} u(a,y)$:*

$$C(x,t) = \left\{ y \in \mathcal{Y} : \ u(\boldsymbol{a}(x,t), y) \geq \boldsymbol{\theta}(x,t) \right\}, \qquad (11)$$

*Further, we have $\nu_{\mathrm{RA}}(C(x,t)) = \boldsymbol{\theta}(x,t)$.*

The optimal sets for RA-CPO (2.2) can now be obtained based on the following re-parametrization in terms of the coverage probabilities that we assign to each $x \in \mathcal{X}$. In order to satisfy the marginal constraint of RA-CPO, we will need to assign to each $x$, a coverage value $t(x)$ such that $\mathbb{E}_X[t(X)] \geq 1 - \alpha$. From the above proposition, if an instance $x$ is assigned with $t$ units of (probability) coverage, then it can add the maximum utility amount of $\boldsymbol{\theta}(x,t)$ to the objective and its corresponding prediction set, which is optimal given $t$ units of coverage assigned to $x$, is given in (11). Hence, to find the optimal prediction sets we should find the assignment $t(x)$ which optimally distributes the $(1-\alpha)$ units of probability over the feature space $\mathcal{X}$, such that the expected utility is optimized. This step is captured by the following equivalent reformulation of RA-CPO:

$$\begin{aligned} \underset{t:\mathcal{X} \to [0,1]}{\text{maximize}} \quad & \mathbb{E}_X\big[\boldsymbol{\theta}(X, t(X))\big] \\ \text{subject to:} \quad & \mathbb{E}_X\big[t(X)\big] \geq 1 - \alpha. \end{aligned} \qquad (12)$$

Once the optimal solution $t^*(x)$ to the above re-parametrization of RA-CPO is found, then the optimal policy/actions, denoted by $a^*(x) = \boldsymbol{a}(x, t^*(x))$, are derived according to (9), and the optimal prediction set is given by:

$$C^*(x) = \left\{ y \in \mathcal{Y} : \ u(a^*(x), y) \geq \boldsymbol{\theta}(x, t^*(x)) \right\}. \qquad (13)$$

Let us summarize what we have done so far: We proved that RA-DPO (2) and RA-CPO (2.2) are equivalent. Then, to solve the RA-CPO we used a reparametrization as in (12), which we will now solve.

Using tools from duality theory, we can show that the optimization problem (12) admits a solution with a simple "one-dimensional" structure in terms of scalar parameter $\beta \in \mathbb{R}$ and an assignment function $\boldsymbol{g} : \mathcal{X} \times \mathbb{R} \to [0,1]$ defined as [3]

$$\boldsymbol{g}(x, \beta) = \arg\max_{s \in [0,1]} \big\{\boldsymbol{\theta}(x,s) + \beta s\big\}. \qquad (14)$$

An illustration of the function $\boldsymbol{g}(x, \beta)$ is provided in Figure 2. One can observe that $\boldsymbol{g}(x, \cdot)$ is connected to the convex-conjugate transform of the function $\boldsymbol{\theta}(x, \cdot)$.

---

[3]For simplicity, we assume in this section that the maximizer of $\theta(x,s) + \beta s$ is unique with probability 1.

**Theorem 3.2.** *Assume that the marginal distribution of $X$ is continuous. Then, optimal solution of* (12) *has the form*

$$t^*(x) = \boldsymbol{g}(x, \beta^*)$$

*for a value $\beta^* \in \mathbb{R}$. Consequently, the optimal prediction sets for RA-CPO are obtained using $t^*(x)$ from* (13). *Further, the value of $\beta^*$ is a solution to the following equation in terms of the scalar $\beta$: $\mathbb{E}_X[\boldsymbol{g}(X, \beta)] = 1 - \alpha$.*

The main implication of the above theorem is that it provides a simple characterization of the optimal sets given access to the data distribution: (i) Find the scalar $\beta^*$ that satisfies $\mathbb{E}_X[\boldsymbol{g}(X, \beta^*)] = 1 - \alpha$; (ii) For each $x \in \mathcal{X}$ compute $t^*(x) := \boldsymbol{g}(x, \beta^*)$ from (14); (iii) The optimal prediction set for $x$, $C^*(x)$, is then given by (13).

The scalar characterization via $\beta$ is particularly useful when only approximate conditional probabilities are available. By substituting $p(y \mid x)$ with an approximation in all definitions, we can still apply Theorem 3.2 to find a $\beta$ that ensures valid coverage for the corresponding prediction sets. This simple scalar calibration then yields prediction sets whose risk-averse utility are improved (and eventually becomes optimal) as the quality of the estimated probabilities improves.

## 4. The Main Algorithm: Risk Averse Calibration (RAC)

In Section 3, we derived the structure of the optimal prediction sets for the RA-CPO problem. These sets are defined by the functions $\boldsymbol{\theta}(x, t)$ and $\boldsymbol{a}(x, t)$ given in (9), which fundamentally relate coverage to utility and actions, as well as the assignment function $\boldsymbol{g}(x, \beta)$ introduced in (14). These quantities are defined based on the true conditional distribution which is often unknown in practice.

In this section, we consider the finite-sample setting. We assume access to calibration samples $\{(X_i, Y_i)\}_{i=1}^n$ and a predictive model $f : \mathcal{X} \to \Delta_{\mathcal{Y}}$, which assigns a $|\mathcal{Y}|$-dimensional probability vector to each $x \in \mathcal{X}$. The output $f_x$ represents approximate label probabilities, such as those from a pre-trained model's softmax layer. We denote by $f_x(y)$ the probability assigned to label $y$ for input $x$.

Using the model $f$, we will estimate the functions $\boldsymbol{\theta}$, $\boldsymbol{a}$, and $\boldsymbol{g}$, defined in (9) and (14), by substituting the true conditional probabilities with their estimated counterparts obtained via $f$. Concretely,

$$\hat{\boldsymbol{\theta}}(x, t) = \max_{a \in \mathcal{A}} \text{quantile}_{1-t}\big[u(a, Y) \mid Y \sim f_x\big], \quad (15)$$

$$\hat{\boldsymbol{a}}(x, t) = \arg\max_{a \in \mathcal{A}} \text{quantile}_{1-t}\big[u(a, Y) \mid Y \sim f_x\big]. \quad (16)$$

and

$$\hat{\boldsymbol{g}}(x, \beta) = \arg\max_{s \in [0,1]} \Big\{ \hat{\boldsymbol{\theta}}(x, s) + \beta s \Big\}. \quad (17)$$

---

**Algorithm 1** Risk Averse Calibration (RAC)

**Input:** Miscoverage level $\alpha$, calibration samples $\{(X_i, Y_i)\}_{i=1}^n$, test covariate $X_{n+1}$.
**For each** $y \in \mathcal{Y}$: solve

$$\hat{\beta}_y = \arg\min_{\beta \in \mathbb{R}} \beta$$

s.t. $\dfrac{1}{n+1}\bigg( \sum_{i=1}^n \mathbf{1}[Y_i \in \hat{C}(X_i; \beta)]$

$$+ \mathbf{1}[y \in \hat{C}(X_{n+1}; \beta)] \bigg) \geq 1 - \alpha$$

**Output:** Compute

$$C_{\text{RAC}}(X_{n+1}) = \big\{ y \in \mathcal{Y} \mid y \in \hat{C}(X_{n+1}; \hat{\beta}_y) \big\}.$$

---

From the result of Theorem 3.2 we know that the optimal prediction sets admit a "one-dimensional" structure in terms of the scalar parameter $\beta \in \mathbb{R}$, and the optimal conditional coverage assignment is derived using the function $\boldsymbol{g}(x, \beta)$. Hence, to simplify notation, we analogously define

$$\hat{\boldsymbol{\theta}}(x, \beta) := \hat{\boldsymbol{\theta}}\big(x, \hat{\boldsymbol{g}}(x, \beta)\big), \quad \hat{\boldsymbol{a}}(x, \beta) := \hat{\boldsymbol{a}}\big(x, \hat{\boldsymbol{g}}(x, \beta)\big).$$

Following (13), the prediction sets take the form

$$\hat{C}(x; \beta) = \Big\{ y \in \mathcal{Y} : \ u\big(\hat{\boldsymbol{a}}(x, \beta), y\big) \geq \hat{\boldsymbol{\theta}}(x, \beta) \Big\}.$$

We can now present our main algorithm.

**Theorem 4.1.** *Assuming that the calibration data $\{(X_i, Y_i)\}_{i=1}^n$ and $(X_{n+1}, Y_{n+1})$ are exchangeable, we have*

$$\Pr\big[Y_{n+1} \in C_{\text{RAC}}(X_{n+1})\big] \geq 1 - \alpha.$$

Put it differently, Theorem 4.1 states that the prediction sets constructed by RAC have the so-called property of distribution-free coverage guarantee. Recalling the definitions (5), we can now state the following corollary.

**Corollary 4.2.** *Assuming that the calibration data $\{(X_i, Y_i)\}_{i=1}^n$ and $(X_{n+1}, Y_{n+1})$ are exchangeable, we have*

$$\Pr\big[u\big(a_{\text{RA}}\left(C_{\text{RAC}}(X_{n+1})\right), Y_{n+1}\big) \geq \nu_{\text{RA}}(C_{\text{RAC}}(X_{n+1}))\big]$$
$$\geq 1 - \alpha.$$

Putting the pieces together, Corollary 4.2 ensures that a simple max-min decision policy over RAC-constructed prediction sets provides a pair of *action policy* and *utility certificate*, namely $a_{\text{RA}}(C_{\text{RAC}}(X_{\text{test}}))$ and $\nu_{\text{RA}}(C_{\text{RAC}}(X_{\text{test}}))$, providing a distribution free safety guarantee according to (1). Moreover, Theorem 3.2 highlights RAC's practical relevance in terms of exploiting the predictive model. Specifically, RAC's utility performance depends on the quality of the predictive model $f$: if $f$ closely estimates the true conditional probabilities, then the model-based definitions in

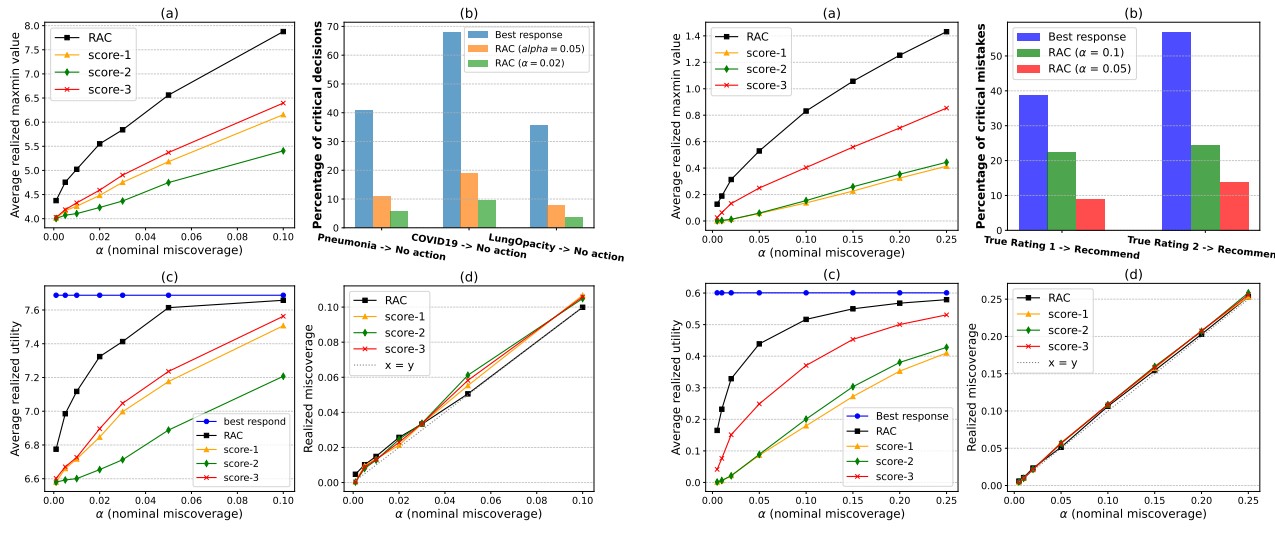

**Medical Diagnosis Experiment**  **MovieLens Recommendation Experiment**

*Figure 3.* Results from two experiments. (a) Average realized max-min value as a function of $\alpha$. (b) Fraction of wrong critical decisions: in medical diagnosis, severe omission of appropriate care (e.g., failing to act on pneumonia or COVID-19 cases); in MovieLens, the percentage of movies rated 1 or 2 that were incorrectly recommended. (c) Average realized utility. (d) Realized miscoverage.

(15) and (17) approximate their true counterparts in (9) and (14), ensuring that RAC-informed decisions align closely with the optimal ones, as guaranteed by Theorem 3.2.

## 5. Experiments

In this section, given a pre-trained model which assigns probability $f_x(y)$ to input-label pair $(x, y)$, we compare RAC with two groups of baselines:

**Calibration + Best-Response.** We calibrate the model on the calibration data using a strengthened version of *decision calibration* (Zhao et al., 2021), specifically the variant from (Noarov et al., 2023), which provides *swap regret* bounds. We then apply the *best-response* policy: best-response$(x)$ $=$ $\arg\max_{a \in \mathcal{A}}$ $\mathbb{E}_{y \sim f_x(y)}\big[u(a, y)\big]$. While this method may achieve higher average utility, it fully trusts the model and is prone to critical errors.

**Conformal Prediction + Max-Min.** We construct $(1 - \alpha)$-valid prediction sets using split conformal prediction with three different scoring rules. The decision policy then applies the max-min rule from Section 2: $a_{\text{RA}}(C(x)) = \arg\max_{a \in \mathcal{A}} \min_{y \in C(x)} u(a, y)$, which we proved is the optimal strategy when deciding based on prediction sets:

- **score-1** (Sadinle et al., 2019): $1 - f_x(y)$,
- **score-2** (Romano et al., 2020): $\sum\limits_{y': f_x(y') > f_x(y)} f_x(y')$,
- **score-3** (Cortes-Gomez et al., 2024): a greedy scoring rule tailored to the max-min policy.

By varying $\alpha$, we can control the degree of conservativeness,

trading off average utility against the avoidance of catastrophic errors. We compare in terms of safety and utility using the following metrics: **(a) Average realized max-min value**: The mean of the worst-case utility across the prediction sets (i.e., the average of $\nu_{\text{RA}}$ in (5)). **(b) Fraction of critical mistakes**: For samples with a critical ground-truth label, we report the fraction of cases in which each method chooses the *worst* action. **(c) Average realized utility**: The empirical mean of the realized utilities across all test samples. **(d) Realized miscoverage**: The fraction of test samples for which the true label is not in the prediction set.

### 5.1. Medical Diagnosis

In this experiment, we explore decision making in medical diagnosis and treatment as a **risk-sensitive** application. We use the *COVID-19 Radiography Database* (Chowdhury et al., 2020; Rahman et al., 2021), containing chest X-ray images of four classes: *Normal*, *Pneumonia*, *COVID-19*, and *Lung Opacity*. The data are randomly split into **training** (70%), **calibration** (10%), and **test** (20%) sets. We then fine-tune an `Inception_v3` model (Szegedy et al., 2015; 2016) (pretrained by google on ImageNet) by retraining the higher layers, while preserving the early-layer features.

To capture clinical priorities, we employ the **utility matrix** in Table 1, which maps each true condition (row) to a set of actions (column). Although we use the specific matrix below, our setup can accommodate any alternative design. (Further details on the *AI-assisted* construction appear in the Appendix C) All the baselines then will be calibrated to connect model's predictions to these four actions.

| True Label | No Action | Antibiotics | Quarantine | Testing |
|---|---|---|---|---|
| Normal | 10 | 2 | 2 | 4 |
| Pneumonia | 0 | 10 | 3 | 7 |
| COVID-19 | 0 | 3 | 10 | 8 |
| Lung Opacity | 1 | 4 | 4 | 10 |

*Table 1.* Utility matrix for the four-class chest X-ray task.

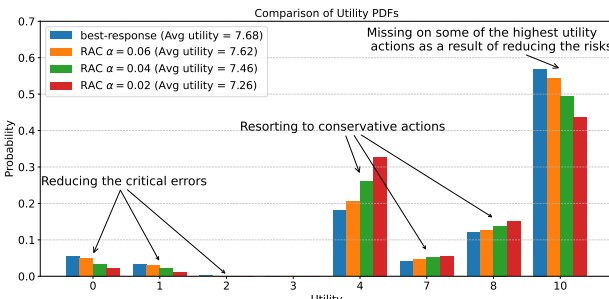

*Figure 4.* Comparison of the full utility distribution for different variants of RAC and the best-response method.

After training, we vary the nominal miscoverage parameter $\alpha$ during calibration to study its impact on performance. As shown in Figure 3(a), our method achieves the best trade-off curve among baselines, providing higher worst-case utilities for every nominal $\alpha$. Equivalently, it offers stronger *utility certificates* at each high-probability threshold. In Figure 3(c), it also consistently outperforms other prediction set-based methods in terms of *average utility*.

While the best-response method attains the highest overall average utility, Figure 3(b) highlights its susceptibility to critical mistakes. For example, in COVID-19 cases, best-response chooses *no action* over 60% of the time, recommending a wrong treatment on a large fraction of patients with COVID-19. Our risk-averse policy (RAC) drive this error rate below 10% (at $\alpha = 0.02$), incurring only a modest (under 5%) drop in average utility. Finally, Figure 3(d) confirms that all prediction-set-based baselines achieve their target miscoverage levels, ensuring the associated high-probability utility guarantees remain statistically valid. Additionally, in Figure 4, we also report the full distribution of the utility of the actions made by RAC for different values of alpha. There, it is even more clear that as we increase $1 - \alpha$, RAC avoids the extremely low utility actions at the cost of missing on some of the highest utility ones, by resorting to conservative decisions.

The plots reported in this experiment can serve a broader purpose beyond evaluation: they provide a practical interface for choosing the right level of risk aversion in real-world deployments. By inspecting trade-offs across different $\alpha$ values, e.g. by looking at plots similar to Figures 3 and 4, practitioners can tune the system to their needs—for instance, favoring safety over utility in high-stakes settings like medicine, or vice versa in lower-risk applications.

**5.2. Recommender Systems**

We next consider a risk-sensitive recommendation scenario using the *MovieLens* dataset. Each data point is a user–movie pair $\left(x = (\text{user features}, \text{movie features}), y\right)$, where the label $y \in \{1, 2, 3, 4, 5\}$ is the user's rating. We split the data into **training** (80%), **calibration** (10%), and **test** (10%), and train a neural network classifier $f$ (details in the Appendix) to estimate the probability distribution $f_y(x)$.

At test time, the policy must decide whether to *recommend* or *not recommend* a movie. We use the **utility function** in

Table 2: if a movie with true rating $y$ is recommended, the utility is $y - 3$, while not recommending yields 0. We vary

| Action | 1 | 2 | 3 | 4 | 5 |
|---|---|---|---|---|---|
| Not Rec | 0 | 0 | 0 | 0 | 0 |
| Rec | -2 | -1 | 0 | +1 | +2 |

*Table 2.* Utility matrix for the MovieLens recommendation task.

the nominal miscoverage $\alpha$ during calibration and measure performance on test data. As shown in Figure 3(a), our method achieves the best trade-off among baselines, offering stronger *utility certificates* (worst-case utility) at all $\alpha$ levels. Figure 3(c) also shows that our approach outperforms other CP-based methods in *average utility*.

Although the best-response method achieves the highest overall average utility, Figure 3(b) reveals its vulnerability to "critical mistakes"—frequently *recommending* movies rated 1 or 2. Such failures can undermine user trust and harm companies policy in keeping their customers. In contrast, RAC ($\alpha = 0.05$) cuts these critical errors by 75%, while incurring only a modest (15%) reduction in average utility.

## 6. Discussion and Future Work

In this paper, we established the decision-theoretic foundations of conformal prediction, showing that valid prediction sets act as sufficient statistics for risk-averse agents optimizing their value at risk. We developed an algorithmic interface linking predictions from any black-box model to actions with marginal, distribution-free safety guarantees.

Although our focus has been primarily on marginal safety guarantees, we acknowledge the practical importance of stronger conditional guarantees. These include group-conditional (based on covariate characteristics), label-conditional (based on true labels), and action-conditional safety (based on chosen actions). Extending our results systematically to these more nuanced scenarios presents promising directions for future research.

## Acknowledgments

This work was supported by the NSF Institute for CORE Emerging Methods in Data Science (EnCORE) and NSF grant FAI-2147212. The authors wish to thank John Cherian, Natalie Collina, and Bruce D. Lee for helpful discussions.

## Impact Statement

We developed a decision theoretic framework for conformal prediction. We do not anticipate negative societal impact.

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

# A. Extended related works

The foundational idea of prediction sets can be traced back to early studies by Wilks (1941); Wald (1943); Scheffe and Tukey (1945); Tukey (1947). The initial concepts of conformal prediction (CP) were introduced in Saunders et al. (1999); Vovk et al. (1999; 2005). With the advancement of machine learning, conformal prediction has become a widely adopted framework for constructing prediction sets (Vovk, 2013; Papadopoulos et al., 2002; Lei et al., 2018; Romano et al., 2020; 2019; Park et al., 2022; Angelopoulos et al., 2020). There has been a growing body of work aiming to adapt conformal prediction methods for a range of decision-making problems. In the following, we will discuss the ones relevant to the present work.

**Risk Control.** A growing line of research extends CP beyond coverage constraints to control more general risk measures (Lindemann et al., 2023; Angelopoulos et al., 2022; 2021; Cortes-Gomez et al., 2024; Lekeufack et al., 2024; Zecchin and Simeone, 2024a; Blot et al., 2024; Zecchin and Simeone, 2024b). In particular, Angelopoulos et al. (2022) propose conformal risk control for monotone risk measures over prediction sets, and Cortes-Gomez et al. (2024) extend this by constructing sets that satisfy coverage while achieving low risk. However, these works do not explicitly discuss which *actions* their sets should inform or how to design these sets to best serve the decision maker. Moreover, Lindemann et al. (2023) applies conformal prediction to safe planning, and Lekeufack et al. (2024) focuses on decisions parameterized by a single scalar, calibrated to control risk. However, these works restrict their action policy to a *predefined* low-dimensional family, leaving open the question of how to *jointly* optimize over policy design and uncertainty quantification for risk-averse utility.

In this paper, we fill this gap by addressing three core questions for a risk-averse decision maker: (1) *What is the correct notion of uncertainty quantification?* We prove that prediction sets are optimal for high-stakes decisions. (2) *How can we design these optimal sets?* We provide an exact population-level characterization and a distribution-free, finite-sample construction. (3) *What is the optimal policy given these sets?* We show that a simple max–min rule is optimal for risk-averse utility. In Section 5, we implement the most recent approach in this direction (Cortes-Gomez et al., 2024), and demonstrate that our framework yields significantly more effective action policies.

On top of the fundamental differences we mentioned, there are also technical differences. After proving the equivalence of the risk-averse objective defined in Section 2 to the prediction set optimization called RA-CPO in Section 2.2, one might think we can define a risk function of the form $l(C) = -\max_{a \in \mathcal{A}} \min_{y \in C(x)} u(a, y)$, and then apply risk controlling methods to control this risk. However, controlling this risk alone is meaningless, as it is always possible to control the risk by outputting trivial sets. Hence, the risk should be controlled combined with coverage guarantees. The only risk controlling framework that additionally allows for a coverage constraint is the work of Cortes-Gomez et al. (2024), which we compare our performance with in Section 5, and show our superior performance in handling the safety utility trade-off. Furthermore, the defined loss function $l$ for a generic utility function $u$, lacks any (approximate) separability property or sub-modularity, which are essential for algorithmic development of Cortes-Gomez et al. (2024). We, however, work directly with the max-min objective and do not rely on any assumptions. For readers familiar with nested conformal prediction (Gupta et al., 2022), perhaps another way to elaborate on this important technical difference is to look at Section **??**, where in Theorem 3.2, we characterize the optimal prediction sets over the population. It is clear then that the optimal sets do *not* necessarily form a nested sequence of sets as we sweep the miscoverage threshold $\alpha$. This is in contrast to when we want to find optimal sets corresponding to minimum average prediction set size (or any other separable objective). There, the optimal characterization is of the form $p(y|x) \geq q$ (or more generally of the form $s(x, y) \leq q$ for some score function $s$), where $q$ is tuned to satisfy the marginal coverage constraint (Lei et al., 2013; Sadinle et al., 2019; Kiyani et al., 2024b). This distinction hints to the sub-optimality of the algorithms that rely on monotonicity properties of the risk, e.g. thresholding a score function, in obtaining the best risk averse action policies and safety guarantee.

**Robust Optimization.** The max-min policy that we will discuss in Section 2.1 also naturally arises at the intersection of uncertainty quantification and robust optimization (Patel et al., 2024b; Johnstone and Cox, 2021; Chenreddy and Delage, 2024; Li et al., 2025; Yeh et al., 2024; Cao, 2024; Wang et al., 2023; Lou et al., 2024; Patel et al., 2024a; Elmachtoub et al., 2023; Lin et al., 2024; Chan et al., 2024; 2023; Chan and Kaw, 2020). In robust optimization, decision-making under uncertainty is typically formulated as a minimax problem, where an optimal decision is sought against worst-case realizations within an uncertainty set. Despite a structural resemblance of these works to our framework in that they involve optimization over an uncertainty set, their scope and objectives have some fundamental differences from ours. We fix any black-box predictive model and any utility function, and in contrast to existing approaches, we *jointly* characterize the optimal notions of uncertainty quantification and action policy. Specifically, we ask: (1) What is the appropriate uncertainty quantification

for risk-averse decision makers? We answer that prediction sets are optimal for achieving high-probability utility guarantees. (2) How should these prediction sets be optimally constructed? We provide a distribution-free, finite-sample construction that characterizes the optimal sets. (3) What is the optimal decision policy given these sets? We prove that the max-min rule is provably optimal for risk averse agents. In doing so, our Risk-Averse Calibration (RAC) method offers a principled alternative to uncertainty sets based on heuristic conformity score designs, thereby contributing to the growing intersection of conformal prediction and robust optimization. Additionally, on a more technical note, in Section **??**, we show that the optimal prediction sets that lead to optimal safe action policies when used in tandem with the max-min rule do *not* necessarily take the form of thresholding a score function (i.e., $s(x, y) \leq q$ for some score function $s$). There, we characterize an alternative form that, in fact, captures the optimal prediction sets in the context of risk-averse decision-making. That is to say, our results hint to a principled alternative to conventional score-based prediction sets in the pipeline of robust optimization to avoid suboptimality.

**Risk Aversion in Economics.** Decision-making under risk aversion is a foundational topic in economics, shaped by seminal contributions. Bernoulli (Bernoulli, 1954) introduced expected utility theory, explaining risk aversion via diminishing marginal utility. Von Neumann and Morgenstern (von Neumann and Morgenstern, 1944) formalized this with an axiomatic model of rational choice under uncertainty. Pratt (Pratt, 1964) and Arrow (Arrow, 1965) developed the Arrow–Pratt coefficients, providing precise measures of risk aversion and distinguishing between increasing and decreasing risk sensitivity. Hadar and Russell (Hadar and Russell, 1969), along with Hanoch and Levy (Hanoch and Levy, 1969), introduced stochastic dominance to rank risky alternatives for risk-averse agents. Rothschild and Stiglitz (Rothschild and Stiglitz, 1970) deepened this framework by defining mean-preserving spreads, a formal notion of increased risk. More recent extensions introduced robust decision-making criteria, such as maximin and minimax-regret, applicable under ambiguous or unknown probabilities (Manski, 2000; 2004; Manski and Tetenov, 2007; Manski, 2011). Collectively, these works established the theoretical underpinnings of risk aversion that continue to influence modern economic theory (for a recent survey look at (Royset, 2024)). In contrast to these works, our work focuses on data-driven learning and uncertainty quantification aspects of the risk averse decision making. We develop distribution-free methods capable of leveraging any black-box pretrained model, accompanied with risk aversion guarantees.

**Further Related Work.** The potential connection of CP ideas to decision making has also been explored in Vovk and Bendtsen (2018), from the point of view of conformal predictive distributions. Conformal predictive distributions produce calibrated distributions rather than prediction sets–see e.g. (Vovk et al., 2017; 2018; 2020). Therefore, they are best to be compared with calibrated forecasts as the methodologies developed in Vovk and Bendtsen (2018) are also targeting expectation maximizer–i.e. risk neutral– agents. Additionally, recent works also explored the application of CP sets in decision making in the context of counterfactual inference (Lei and Candès, 2021; Yin et al., 2024; Jin et al., 2023). We, however, focus on risk averse decision making using prediction sets. In particular, we show that prediction sets are a sufficient statistic for risk averse agents that aim to optimize their value at risk.

Alternatively, Bayesian methods for risk-averse decision-making often employ Gaussian Processes (GPs) to optimize measures like Value-at-Risk and Conditional Value-at-Risk; e.g. look at (Sui et al., 2015; Nguyen et al., 2021; Demirel et al., 2022; Cakmak et al., 2020; Lin et al., 2022; Baudry et al., 2021; Jia et al., 2024). These approaches rely on accurate Bayesian posterior distributions, thus implicitly assuming well-specified probabilistic models. Our conformal approach complements rather than competes with Bayesian methods: our theoretical results (up to Section **??**) can be directly employed even in Bayesian settings. In fact, when Bayesian approximations are reliable, one can take advantage of our optimal prediction sets derivation in population, and then calibrate the prediction sets with finite sample under Bayesian models, without employing the finite-sample calibration of Section 4. Alternatively, even when Bayesian assumptions' precision is uncertain, one can still start from Bayesian posteriors and further calibrate prediction sets using our approach, ensuring robust safety guarantees.

Although our primary aim is to develop a general framework to construct prediction sets for high-stakes decision-making, we note that conformal prediction sets have been explored in a wide range of specific applications and domains of high-stakes nature. For instance, CP methods have been adapted and used in medical tasks (Banerji et al., 2023), power and energy systems (Renkema et al., 2024), formal verification and control (Lindemann et al., 2024), chance-constrained optimization (Zhao et al., 2024), and more generally Sun et al. (2024); Ramalingam et al. (2024); Kiyani et al. (2024a); Straitouri et al. (2023); Vishwakarma et al. (2024); Kiyani et al. (2024b); van der Laan and Alaa (2024); Noorani et al. (2024). Our framework could potentially be extended to these domains, yet each may present additional, domain-specific challenges that lie beyond the scope of this work.

# B. Proofs

## B.1. Proof of Proposition 2.2

We prove that the risk-averse decision rule

$$a_{\mathrm{RA}}\big(C(x)\big) \ := \ \arg\max_{a \in \mathcal{A}} \min_{y \in C(x)} u(a, y)$$

solves the minimax problem in (7).

**Part 1: Upper bound for any arbitrary policy.** Let $\pi(\cdot) : 2^{\mathcal{Y}} \to \mathcal{A}$ be any policy, and let $C(\cdot)$ be a fixed set function satisfying

$$\Pr_{(X,Y)\sim\mathcal{P}}\big[Y \in C(X)\big] \ \geq \ 1 - \alpha.$$

We construct a "worst-case" distribution in $\Omega$ for $\pi$.

Pick any $x \in \mathcal{X}$ for which $C(x) \neq \emptyset$. Define a distribution $p^*(x, y)$ by

$$p^*(X = x) \ = \ 1, \quad p^*(Y = y \mid X = x) \ = \ \begin{cases} 1 & \text{for some } y \in \arg\min_{z \in C(x)} u\big(\pi(C(x)), z\big), \\ 0 & \text{otherwise.} \end{cases}$$

Under $p^*$, we have $Y \in C(X)$ almost surely (since $C(x)$ is nonempty and we place all mass on a label in $C(x)$). Hence $p^* \in \Omega$ because the marginal coverage constraint

$$\Pr_{(X,Y)\sim p^*}[Y \in C(X)] = 1 \ \geq \ 1 - \alpha$$

is satisfied. But under this distribution, the utility of $\pi(C(x))$ is forced to be

$$\min_{y \in C(x)} u\big(\pi(C(x)), y\big),$$

since $Y$ is chosen (with probability 1) to be the worst-case label within $C(x)$. Thus, for this specific $x$, no matter how we choose $\pi$, its achievable value is at most $\min_{y \in C(x)} u\big(\pi(C(x)), y\big)$. Also,

$$\min_{y \in C(x)} u\big(\pi(C(x)), y\big) \leq \max_{a \in \mathcal{A}} \min_{y \in C(x)} u\big(a, y\big),$$

Because $x$ was arbitrary (among those with $C(x) \neq \emptyset$), repeating the same argument for each such $x$ yields

$$\inf_{p \in \Omega} \nu^*(\pi, p) \ \leq \ \inf_{x \, : \, C(x) \neq \emptyset} \max_{a \in \mathcal{A}} \min_{y \in C(x)} u\big(a, y\big).$$

In other words, *any* policy $\pi$ cannot achieve a value larger than the above infimum for the inner minimization in (7).

**Part 2: Achievability by the** $\max\min$ **policy.** Next, we show that the policy

$$\pi^*\big(C(x)\big) \ = \ \arg\max_{a \in \mathcal{A}} \min_{y \in C(x)} u(a, y)$$

matches the upper bound from Part 1 and is thus minimax optimal. Consider any $p \in \Omega$.

Define

$$\nu(x) \ := \ \max_{a \in \mathcal{A}} \min_{y \in C(x)} u(a, y).$$

For those $x \in \mathcal{X}$ such that $C(x)$ is empty put $\nu(x) = \max_{a \in \mathcal{A}} \max_{y \in \mathcal{Y}} u(a, y)$. We claim that with probability at least $1 - \alpha$, the policy $a_{\mathrm{RA}}(C(x))$ achieves a utility at least $\nu(x)$. Indeed, on the event $\{Y \in C(X)\}$ (which has probability at least $1 - \alpha$ by assumption), it holds that

$$u\Big(a_{\mathrm{RA}}\big(C(X)\big), Y\Big) \ \geq \ \min_{y \in C(X)} u\Big(a_{\mathrm{RA}}\big(C(X)\big), y\Big) \ = \ \nu(X).$$

Thus, setting the target utility at each $x$ to $\nu(x)$ satisfies

$$\Pr_{(X,Y)\sim p}\Big[u\big(a_{\mathrm{RA}}(C(X)),Y\big) \geq \nu(X)\Big] \geq 1-\alpha.$$

By definition of $\nu^*(\cdot,\cdot)$, this implies

$$\nu^*\big(a_{\mathrm{RA}},p\big) \geq \mathbb{E}_{X\sim p}\big[\nu(X)\big] = \mathbb{E}_{X\sim p}\Big[\max_a \min_{y\in C(X)} u(a,y)\Big] \geq \inf_{x\,:\,C(x)\neq\emptyset} \max_{a\in\mathcal{A}} \min_{y\in C(x)} u\big(a,y\big).$$

Since $p \in \Omega$ was arbitrary, we have shown

$$\inf_{p\in\Omega} \nu^*\big(a_{\mathrm{RA}},p\big) \geq \inf_{x\,:\,C(x)\neq\emptyset} \max_{a\in\mathcal{A}} \min_{y\in C(x)} u\big(a,y\big).$$

Comparing with the upper bound in Part 1 establishes that $a_{\mathrm{RA}}$ attains the best possible (minimax) value. Hence

$$\pi^*(x) = a_{\mathrm{RA}}\big(C(x)\big) = \arg\max_{a\in\mathcal{A}} \min_{y\in C(x)} u(a,y)$$

solves the minimax problem (7).

### B.2. Proof of Theorem 2.3

We give a constructive proof by showing how from each solution of RA-DPO we can construct a feasible solution of RA-CPO without losing any utility, and vice versa. By applying this to the optimal solutions of both problems, we obtain the result of the theorem.

**(I) From RA-DPO to RA-CPO.** Suppose we have an feasible solution $\big(a(\cdot),\nu(\cdot)\big)$ to the RA-DPO problem. Consider a pair $(a(\cdot),\nu(\cdot))$ such that $a : \mathcal{X} \to \mathcal{A}$ and $\nu : \mathcal{X} \to [0,u_{\max}]$. Here, we have $u_{\max} = max_a \max_y u(a,y)$, and as mentioned in Section 2, since $\nu$ is a utility certificate its value at any $x$ should be less than $u_{\max}$. Since $(a,\nu)$ is a feasible solution of RA-DPO, it satisfies the following:

$$\Pr_{X,Y}\left[u(a(X),Y) \geq v(X)\right] \geq 1-\alpha.$$

Define a prediction set

$$C(x) = \Big\{ y \mid u\big(a(x),y\big) \geq \nu(x)\Big\}. \tag{18}$$

In words, $C(x)$ is the set of labels $y$ for which the utility $u\big(a(x),y\big)$ is at least $\nu(x)$. By definition, we have

$$\Pr\big[Y \in C(X) \mid X = x\big] = \Pr\big[u\big(a(X),Y\big) \geq \nu(X) \mid X = x\big].$$

As a result, we have

$$\begin{aligned}
\Pr\big[Y \in C(X)\big] &= \mathbb{E}_X\big[\Pr\big[Y \in C(X) \mid X\big]\big] \\
&= \mathbb{E}_X\big[\Pr\big[u(a(X),Y) \geq \nu(X) \mid X\big]\big] \\
&= \Pr\big[u(a(X),Y) \geq \nu(X)\big] \\
&\geq 1-\alpha.
\end{aligned}$$

Hence, $C(\cdot)$ satisfies the marginal coverage constraint of RA-CPO.

Next, we will improve the prediction sets $C$ to new prediction sets $\tilde{C}$ which satisfy the marginal guarantee but can potentially have larger value under the objective of RA-CPO. The basic idea is to consider points $x \in \mathcal{X}$ such that $C(x)$ is empty and augment an additional element to those empty sets. Recall that we defined $u_{\max} := \max_{a\in\mathcal{A}} \max_{y\in\mathcal{Y}} u(a,y)$. Hence, there exists at least one (action, label) pair, which we call $(a_{\max},y_{\max})$ such that $u_{\max} = u(a_{\max},y_{\max})$. Now, let us define

$$\mathcal{X}_{\mathrm{empty}} = \{x \in \mathcal{X} : C(x) = \emptyset\},$$

where $\emptyset$ denotes the empty set. We now update $C(\cdot)$ to $\tilde{C}(\cdot)$ as follows:

$$\begin{aligned}
\text{- if } x \in \mathcal{X}_{\mathrm{empty}} : &\quad \tilde{C}(x) = \{y_{\max}\}, \\
\text{- if } x \notin \mathcal{X}_{\mathrm{empty}} : &\quad \tilde{C}(x) = C(x).
\end{aligned}$$

Note that we have for any $x \in \mathcal{X}$ that $C(x) \subseteq \tilde{C}(x)$,. Hence, $\tilde{C}(\cdot)$ satisfies the marginal coverage guarantee as $C(\cdot)$ is marginally valid.

Next, we show that the RA-CPO objective under $\tilde{C}(\cdot)$ is at least equal to the RA-DPO objective under $\big(a(\cdot), \nu(\cdot)\big)$. Recall that the RA-CPO objective evaluated at $\tilde{C}(\cdot)$ is

$$\mathbb{E}_X\Big[\max_{a \in \mathcal{A}} \min_{y \in \tilde{C}(X)} u(a, y)\Big].$$

To bound this objective value, we consider two cases based on whether of not $x$ belongs to $\mathcal{X}_{\text{empty}}$.

Consider first the case $x \notin \mathcal{X}_{\text{empty}}$. By definition of $C(x)$ from (18), we have $\min_{y \in C(x)} u(a(x), y) \geq \nu(x)$. Hence, for $x \notin \mathcal{X}_{\text{empty}}$, by noting that $C(x) \neq \emptyset$, we have

$$\max_{a \in \mathcal{A}} \min_{y \in C(x)} u(a, y) \geq \min_{y \in C(x)} u\big(a(x), y\big) \geq \nu(x).$$

Therefore, for $x \notin \mathcal{X}_{\text{empty}}$, by noting that $\tilde{C}(x) = C(x)$, we have

$$\max_{a \in \mathcal{A}} \min_{y \in \tilde{C}(x)} u(a, y) = \max_{a \in \mathcal{A}} \min_{y \in C(x)} u(a, y) \geq \nu(x).$$

Now, let's consider the other case where $x \in \mathcal{X}_{\text{empty}}$. For this case, we not that as $C(x) = \{y_{\max}\}$, and from the fact that for any $x \in \mathcal{X}$ we have $\nu(x) \leq u_{\max}$, we can simply derive

$$\max_{a \in \mathcal{A}} \min_{y \in \tilde{C}(x)} u(a, y) = u_{\max} \geq \nu(x).$$

Therefore, putting the two cases above together, we have proven

$$\mathbb{E}_X\Big[\max_{a \in \mathcal{A}} \min_{y \in \tilde{C}(X)} u(a, y)\Big] \geq \mathbb{E}_X\big[\nu(X)\big]$$

Hence, we have constructed a feasible solution to RA-CPO, namely $\tilde{C}(\cdot)$, that achieves an objective value for RA-CPO which is at least as big as the value of RA-DPO achieved by $(a(\cdot), \nu(\cdot))$. Thus, starting from an a solution of RA-DPO, we have constructed a solution to RA-CPO with at least the same objective value.

**(II) From RA-CPO to RA-DPO.** Conversely, suppose we have a feasible solution $C(\cdot)$ to RA-CPO, which is marginally valid, i.e.

$$\Pr\big[Y \in C(X)\big] \geq 1 - \alpha$$

Define a the action policy $a(\cdot)$ and utility certificate $\nu(\cdot)$ as follows:

$$a(x) := \arg\max_{a \in \mathcal{A}} \min_{y \in C(x)} u(a, y), \quad \text{and} \quad \nu(x) = \max_{a \in \mathcal{A}} \min_{y \in C(x)} u(a, y).$$

It is now easy to see that

$$\Pr\big[u(a(X), Y) \geq \nu(X)\big] = \Pr\big[Y \in C(X)\big] \geq 1 - \alpha.$$

Moreover, by definition of $\nu(x)$, we can easily deduce

$$\mathbb{E}_X[\nu(X)] = \mathbb{E}_X[\max_{a \in \mathcal{A}} \min_{y \in C(X)} u(a, y)].$$

Thus, from a feasible solution of RA-CPO, we constructed a feasible solution to RA-DPO that attains the same objective value, proving the equivalence in the other direction.

### B.3. Proof of Proposition 3.1

*Proof of Proposition 3.1.* Fix any instance $x \in \mathcal{X}$ and a coverage value $t \in [0, 1]$. Recall from (9) that

$$\boldsymbol{\theta}(x, t) = \max_{a \in \mathcal{A}} \text{quantile}_{1-t}\big[u(a, Y) \mid X = x\big], \quad \boldsymbol{a}(x, t) = \arg\max_{a \in \mathcal{A}} \text{quantile}_{1-t}\big[u(a, Y) \mid X = x\big].$$

We want to show that among all sets $C$ with $\Pr[Y \in C \mid X = x] \geq t$, the set

$$C(x, t) = \big\{ y \in \mathcal{Y} : u\big(\boldsymbol{a}(x, t), y\big) \geq \boldsymbol{\theta}(x, t) \big\}$$

maximizes the risk-averse utility $\nu_{\text{RA}}(C) = \max_{a \in \mathcal{A}} \min_{y \in C} u(a, y)$, and the maximum value is $\boldsymbol{\theta}(x, t)$.

**Step 1: Any set $C$ with coverage $\geq t$ has risk-averse utility at most $\boldsymbol{\theta}(x,t)$.** Take an arbitrary set $C \subseteq \mathcal{Y}$ satisfying

$$\Pr\big[Y \in C \mid X = x\big] \;\geq\; t.$$

Then for any action $a \in \mathcal{A}$,

$$\min_{y \in C} u(a,y) \;\leq\; \text{quantile}_{1-t}\big[u(a,Y) \mid X = x\big].$$

(The reason is that with probability at least $t$, $Y$ lies in $C$, and so the $(1-t)$-quantile of $u(a,Y)$ cannot be smaller than the smallest utility on this event.) Taking the maximum over $a$ yields

$$\max_{a \in \mathcal{A}} \min_{y \in C} u(a,y) \;\leq\; \max_{a \in \mathcal{A}} \text{quantile}_{1-t}\big[u(a,Y) \mid X = x\big] \;=\; \boldsymbol{\theta}(x,t).$$

Hence no set with coverage at least $t$ can achieve risk-averse utility larger than $\boldsymbol{\theta}(x,t)$.

**Step 2: The set $C(x,t)$ attains coverage $t$ and achieves $\boldsymbol{\theta}(x,t)$.** Consider $C(x,t) = \{\, y : u(\boldsymbol{a}(x,t),y) \geq \boldsymbol{\theta}(x,t)\,\}$. By definition of the $(1-t)$-quantile, we have

$$\Pr\big[u(\boldsymbol{a}(x,t),Y) \;\geq\; \boldsymbol{\theta}(x,t) \mid X = x\big] \;\geq\; t,$$

which implies $\Pr[Y \in C(x,t) \mid X = x] \geq t$. Moreover, for every $y \in C(x,t)$, by construction

$$u\big(\boldsymbol{a}(x,t),y\big) \;\geq\; \boldsymbol{\theta}(x,t),$$

so

$$\min_{y \in C(x,t)} u\big(\boldsymbol{a}(x,t),y\big) \;\geq\; \boldsymbol{\theta}(x,t).$$

Thus

$$\nu_{\text{RA}}(C(x,t)) \;=\; \max_{a \in \mathcal{A}} \min_{y \in C(x,t)} u(a,y) \;\geq\; \min_{y \in C(x,t)} u\big(\boldsymbol{a}(x,t),y\big) \;\geq\; \boldsymbol{\theta}(x,t).$$

Combining both steps shows that $C(x,t)$ is an optimal choice among all sets with coverage at least $t$, and its risk-averse utility equals $\boldsymbol{\theta}(x,t)$. $\qquad\square$

### B.4. Proof of Theorem 3.2

We start from the reparametrization of RA-CPO given in (12):

$$\begin{aligned}
\underset{t:\mathcal{X}\to[0,1]}{\text{maximize}} \quad & \mathbb{E}_X\big[\boldsymbol{\theta}(X,t(X))\big] \\
\text{subject to:} \quad & \mathbb{E}_X\big[t(X)\big] \;\geq\; 1-\alpha.
\end{aligned}$$
(Reparametrization of RA-CPO)

We will further reparametrize this optimization problem and find equivalent relaxations. To do so, let us define

$$\rho(x,t) = \mathbf{1}[t \leq t(x)]. \tag{19}$$

Also, we will need to consider the derivative of the function $\boldsymbol{\theta}(x,t)$ in terms of its second argument $t$. Since the function $\boldsymbol{\theta}$ can be discontinuous, we will have to consider its generalized derivative (i.e. consider delta functions). More precisely, let $\boldsymbol{\theta}'(x,.) : \mathbb{R} \to \mathbb{R}^*$ where $\mathbb{R}^*$ is the space of functionals on $\mathbb{R}$, such that $\boldsymbol{\theta}'(x,.)$ is the generalized derivative of $\boldsymbol{\theta}(x,.)$. In other words, for any real values $a$ and $b$,

$$\int_a^b \boldsymbol{\theta}'(x,t)dt = \boldsymbol{\theta}(x,b) - \boldsymbol{\theta}(x,a).$$

We can just think of $\boldsymbol{\theta}'(x,t)$ as the derivative $\frac{d}{dt}\boldsymbol{\theta}(x,t)$. We can then rewrite the objective of our optimization problem as

$$\mathbb{E}_X\left[\boldsymbol{\theta}(X,t)\right] = u_{\max} + \mathbb{E}_X \int_{t=0}^1 \rho(X,t)\boldsymbol{\theta}'(X,t)dt,$$

where we used the fact that $\boldsymbol{\theta}(x,0) = u_{\max}$ for any $x \in \mathcal{X}$ by definition and $\boldsymbol{\theta}(x,t) - \boldsymbol{\theta}(x,0) = \int_0^t \boldsymbol{\theta}'(x,t)dt$. Similarly, we can rewrite the constraint as,

$$\mathbb{E}_X\left[t(X)\right] = \mathbb{E}_X \int_{t=0}^1 \rho(X,t)dt.$$

Given the above notation and relations, we can write down the following equivalent reparametrization of (Reparametrization of RA-CPO). The optimization variable here is the function $\rho(x,t)$ which is a step function according to (19). We further note that any such step function defined on the unit interval can be equivalently thought of as a non-increasing function on the unit interval which only takes its value in the set $\{0,1\}$. Hence we arrive at the following integer program that is an equivalent reparametrization of (Reparametrization of RA-CPO) as well as the RA-CPO.

$$
\begin{aligned}
\underset{\substack{\rho(x,t)\in\{0,1\} \\ \forall x\in\mathcal{X}, t\in[0,1]}}{\text{maximize}} \quad & \int_{\mathcal{X}}\int_{t=0}^1 \rho(x,t)p(x)\boldsymbol{\theta}'(x,t)dxdt \\
\text{subject to:} \quad & \int_{\mathcal{X}}\int_{t=0}^1 p(x)\rho(x,t)dxdt \geq 1-\alpha \\
& \rho(x,t) = \text{non-increasing in } t
\end{aligned}
\tag{Integer Program}
$$

We now consider a relaxation of the above integer program to the following convex program. As we will see later, this relaxation becomes equivalent to the above integer program as every solution of the relaxed program would correspond to a solution of the integer program. However, for now, let us focus on the following continuous relaxation whose variable $\rho(x,t)$ can take values in the interval $[0,1]$ (in contrast to the original integer program in which $\rho$ could take its value only in the set $\{0,1\}$):

$$
\begin{aligned}
\text{maximize} \quad & \int_{\mathcal{X}}\int_{t=0}^1 \rho(x,t)p(x)\boldsymbol{\theta}'(x,t)dxdt \\
\text{subject to:} \quad & \int_{\mathcal{X}}\int_{t=0}^1 p(x)\rho(x,t)dxdt \geq 1-\alpha \\
& \rho(x,t) \in [0,1] \quad \forall x \in \mathcal{X}, t \in [0,1] \\
& \rho(x,t) = \text{non-increasing in } t
\end{aligned}
\tag{Relaxed Program}
$$

Here, the "optimization variable" $\rho(x,t)$ belongs to an infinite-dimensional space. Hence, in order to be fully rigorous, we will need to use the duality theory developed for general linear spaces that are not necessarily finite-dimensional. For a reader who is less familiar with infinite-dimensional spaces, what appears below is a direct extension of the duality theory (i.e. writing the Lagrangian) for the usual linear programs in finite-dimensional spaces.

Let $\mathcal{F}$ be the set of all measurable function defined on $\mathcal{X} \times [0,1]$. Note that $\mathcal{F}$ is a linear space. Let $\Omega$ be the set of all the measurable functions on $\mathcal{X} \times [0,1]$ which are non-increasing in $t$ and are bounded between 0 and 1; I.e.

$$\Omega = \{\rho \in \mathcal{F} \text{ s.t. } \rho : \mathcal{X} \times [0,1] \to [0,1]; \forall x \in \mathcal{X} : \rho(x,t) \text{ is non-increasing in } t\} \tag{20}$$

Note that $\Omega$ is a convex set. We can then rewrite the (Relaxed Program) as follows:

$$
\begin{aligned}
\text{maximize} \quad & \int_{\mathcal{X}}\int_{t=0}^1 \rho(x,t)p(x)\boldsymbol{\theta}'(x,t)dxdt \\
\text{subject to:} \quad & \int_{\mathcal{X}}\int_{t=0}^1 p(x)\rho(x,t)dxdt - (1-\alpha) \geq 0 \\
& \rho \in \Omega
\end{aligned}
$$

Moreover, let us define the functional $F : \mathcal{F} \to \mathbb{R}$ as

$$F(\rho) = \int_{\mathcal{X}\times[0,1]} \rho(x,t)p(x)\boldsymbol{\theta}'(x,t)dxdy, \tag{21}$$

and also define the functional $G : \mathcal{F} \to \mathbb{R}$ as

$$G(\rho) = \int_{\mathcal{X}\times[0,1]} \rho(x,t)p(x)dxdt - (1-\alpha). \tag{22}$$

Using the above-defined notation, our program becomes:

$$\begin{aligned} \text{maximize} \quad & F(\rho) \\ \text{subject to:} \quad & G(\rho) \geq 0 \\ & \rho \in \Omega \end{aligned}$$

Note that the feasibility set of the above program is non-empty, as $\rho(x,t) = 1 - \alpha$, for all $(x,t) \in \mathcal{X} \times [0,1]$, is a feasible point. We can now use the duality theory of convex programs in vector spaces (See Theorem 1, Section 8.3 of (Luenberger, 1969)). Specifically, let OPT be the optimal value achievable in the above linear program. Then, there exists a scalar $\beta \geq 0$ such that the following holds:

$$\text{OPT} = \sup_{\rho \in \Omega} \left\{ F(\rho) + \beta G(\rho) \right\}, \tag{23}$$

Here, note that $\beta$ is the usual Lagrange multiplier.

By using (21), in order to solve the optimization in (23) we need to solve the following optimization (note that we change $\inf$ to $\sup$ by applying a negative sign):

$$\sup_{\rho \in \Omega} \left\{ \int_{\mathcal{X} \times [0,1]} p(x)\rho(x,t) \left( \boldsymbol{\theta}'(x,t) + \beta \right) dx dt \right\} + \beta(1 - \alpha).$$

We denote the optimal solution of the above optimization problem by $\rho_\beta^*(x,t)$. From the above optimization problem, it is clear that the optimal solution can be determined individually for every $x \in \mathcal{X}$. We will use Lemma B.1, provided below, to characterize the optimizer of the above optimization. From the lemma, and assuming that, almost surely for every $x \in \mathcal{X}$, the maximizer of $\theta(x,t) + \beta t$ is unique over $t$, we obtain:

$$\rho_\beta^*(x,t) = \mathbf{1}\{t \leq t^*(x)\}, \tag{24}$$

where

$$t^*(x) = \arg \max_{s \in [0,1]} \int_{t=0}^{s} \left( \boldsymbol{\theta}(x,t) + \beta \right) dt = \arg \max_{s \in [0,1]} \left\{ \boldsymbol{\theta}(x,s)dt + \beta s \right\} := \boldsymbol{g}(x, \beta).$$

And the value of $\beta$ should then be chosen such that this optimal solution satisfies the coverage constraint.

We finally note that the optimal solution $\rho_\beta(x,t)$ given in (24) is integer valued. As a result, there is a zero relaxation gap from the (Integer Program) to the (Relaxed Program).

**Lemma B.1.** *Let $\theta : [0,1] \to \mathbb{R}$. Also, let $\Omega$ be the set of all the integrable functions $\rho : [0,1] \to [0,1]$ which are non-decreasing. Consider the following optimization problem:*

$$\max_{\rho \in \Omega} \int_0^1 \theta'(t)\rho(t)dt,$$

*where $\theta'$ denotes the derivative of $\theta$ with respect to $t$ – i.e., $\theta(a) - \theta(b) = \int_a^b \theta'(t)dt$. Then, the the of solutions of the above optimization problem consists of functions $\rho^*$ such that*

$$\rho^*(t) \in \text{ConvexHull}\left( \left\{ \mathbf{1}[t \leq t^*]; \quad t^* \in \arg \max_{t \in [0,1]} \theta(t) \right\} \right),$$

*is a solution to the above optimization problem. As a corollary, if $\theta$ has a unique maximizer $t^*$, then its corresponding $\rho^*(t) = \mathbf{1}[t \leq t^*]$ is the unique solution of the above optimization problem.*

*Proof.* For every $\rho \in \Omega$ write using integration by parts:

$$\int_0^1 \rho(t)\theta'(t)dt = \rho(1)\theta(1) - \rho(0)\theta(0) - \int_0^1 \rho'(t)\theta(t)dt$$

Let us define $\theta_{\max} := \max_{t \in [0,1]} \theta(t)$. Since $\rho(t)$ is a non-increasing function, we have

$$-\int_0^1 \rho'(t)\theta(t)dt \overset{(a)}{\leq} -\theta_{\max}\int_0^1 \rho'(t)dt = \theta_{\max}(\rho(0) - \rho(1)),$$

where the step (a) is obtained because $-\rho(t)$ is non-negative. As a result, we obtain

$$
\begin{aligned}
\int_0^1 \rho(t)\theta'(t)dt &\leq \theta_{\max}\left(\rho(0) - \rho(1)\right) + \rho(1)\theta(1) - \rho(0)\theta(0) \\
&= \left(\theta_{\max} - \theta(0)\right)\left(\rho(0) - \rho(1)\right) + \rho(1)\left(\theta(1) - \theta(0)\right) \\
&\overset{(b)}{\leq} \theta_{\max} - \theta(0),
\end{aligned}
$$

where step (b) is obtained since $\rho(0), \rho(1) \in [0,1]$ and $\theta(1) \leq \theta_{\max}$.

Now, it it easy to see that if $t^*$ is such that $\theta(t^*) = \theta_{\max}$ then both steps (a), (b) will be equality (instead of an inequality) for the following function

$$\rho^*(t) = \mathbf{1}[t \leq t^*].$$

On the other hand, for step (a) to be tight we must have the following: For every point $t$ such that $\rho'(t) < 0$, we have $\theta(t) = \theta_{\max}$. This shows that an optimal solution must be in the convex hull defined in the theorem, and hence, the result of the theorem follows. The uniqueness also follows similarly. $\qquad \square$

### B.5. Proof of Theorem 4.1

We have:

$$
\begin{aligned}
\Pr\left[Y_{n+1} \in C_{\mathrm{RAC}}(X_{n+1})\right] &\overset{(a)}{=} \Pr\left[Y_{n+1} \in \hat{C}\left(X_{n+1}; \hat{\beta}_{Y_{n+1}}\right)\right] \\
&= \mathbb{E}[\mathbf{1}[Y_{n+1} \in \hat{C}\left(X_{n+1}; \hat{\beta}_{Y_{n+1}}\right)]] \\
&\overset{(b)}{=} \mathbb{E}\left[\frac{1}{n+1}\left(\sum_{i=1}^n \mathbf{1}[Y_i \in \hat{C}\left(X_i; \hat{\beta}_{Y_{n+1}}\right)] + \mathbf{1}[Y_{n+1} \in \hat{C}\left(X_{n+1}; \hat{\beta}_{Y_{n+1}}\right)]\right)\right] \\
&\overset{(c)}{\geq} 1 - \alpha.
\end{aligned}
\tag{25}
$$

where, (a) comes form the definition of the prediction set. (b) comes from the fact that

$$\left\{\left(X_1, Y_1, \hat{\beta}_{Y_{n+1}}\right), \ldots, \left(X_n, Y_n, \hat{\beta}_{Y_{n+1}}\right), \left(X_{n+1}, Y_{n+1}, \hat{\beta}_{Y_{n+1}}\right)\right\}$$

are exchangeable, which is due to the fact that (i) the exchangeability of the original $(n+1)$ pairs $\{(X_i, Y_i)\} \cup \{(X_{n+1}, Y_{n+1})\}$, and (ii) the symmetric way in which Algorithm 1 assigns $\hat{\beta}_y$ to each $y \in \mathcal{Y}$. Finally, (c) follows from the definition of $\hat{\beta}_{Y_{n+1}}$.

## C. Utility function for medical experiment

Our results and findings in the medical experiment of section 5.1, can be reproduced with any other reasonable design of utility function. The goal of that experiment is not to capture a precise characterization of difficulties and consequences in medical decision making but rather to pinpoint the advantages of a risk averse calibration approach in sensitive tasks like medical decision making. Of course, in real world scenarios, a more comprehensive approach is needed to define a principled utility function that captures the interests of all the involving parties. That being said, for the sake of proof of concept, we designed a utility matrix using the ChatGPT o1 model by OpenAI. The following is an AI generated text justifying the proposed utility matrix.

**Clinical Justification of the Utility Matrix**

The utility matrix presented in Table 3 reflects the balance of benefits and harms associated with different medical actions for each true clinical condition. Each utility value is determined based on standard clinical guidelines and evidence-based practices, ensuring that the chosen actions optimize patient outcomes while minimizing potential risks.

**Normal (No Disease)**

- **No Action = 10**
  For a patient who is truly healthy, no intervention is optimal as it avoids unnecessary costs, side effects, and patient anxiety. Unwarranted use of antibiotics or quarantine measures can lead to adverse effects and resource wastage (NIHCE, 2015).

- **Antibiotics = 2**
  Prescribing antibiotics to a healthy individual can contribute to antimicrobial resistance and cause side effects without any clinical benefit (CDC, 2022).

- **Quarantine = 2**
  Quarantining a healthy person imposes unnecessary social and psychological burdens without providing any medical advantage (CDC, 2020).

- **Testing = 4**
  While testing can confirm the absence of disease, routine testing in healthy individuals is often not cost-effective and may lead to unnecessary follow-up procedures (of Radiology, 2023).

**Pneumonia**

- **No Action = 0**
  Untreated pneumonia can lead to rapid deterioration and increased mortality, making inaction highly detrimental (Metlay et al., 2019).

- **Antibiotics = 10**
  Timely administration of appropriate antibiotics is crucial for treating bacterial pneumonia, improving survival rates and reducing complications (Metlay et al., 2019; WHO, 2021).

- **Quarantine = 3**
  While some forms of pneumonia may be contagious, standard infection control measures are generally more beneficial than full quarantine, especially when bacterial pneumonia is suspected (Metlay et al., 2019).

- **Testing = 7**
  Diagnostic tests such as chest imaging and sputum cultures are essential for confirming pneumonia and guiding antibiotic therapy (Metlay et al., 2019).

**COVID-19**

- **No Action = 0**
  Ignoring a COVID-19 infection can result in severe disease progression and widespread transmission, making inaction extremely harmful (CDC, 2020).

- **Antibiotics = 3**
  Since COVID-19 is viral, antibiotics are generally only useful if there is a suspected secondary bacterial infection (WHO, 2020).

- **Quarantine = 10**
  Quarantining individuals with COVID-19 is essential for controlling the spread of the virus and protecting public health (CDC, 2020).

- **Testing = 8**
  Confirmatory testing is vital for diagnosing COVID-19 and guiding appropriate interventions, including quarantine and specific therapies (CDC, 2020).

| True Label | No Action | Antibiotics | Quarantine | Testing |
|---|---|---|---|---|
| Normal | 10 | 2 | 2 | 4 |
| Pneumonia | 0 | 10 | 3 | 7 |
| COVID-19 | 0 | 3 | 10 | 8 |
| Lung Opacity | 1 | 4 | 4 | 10 |

*Table 3.* Utility matrix for the four-class chest X-ray task.

## Lung Opacity

- **No Action = 1**
  Ignoring lung opacities can lead to missed diagnoses of serious conditions such as malignancies or tuberculosis, posing significant risks (Rubin et al., 2020).

- **Antibiotics = 4**
  Empirical antibiotic therapy may be beneficial if an infectious etiology is suspected, but it is not universally appropriate and may lead to resistance (CDC, 2022; Metlay et al., 2019).

- **Quarantine = 4**
  Quarantine may be necessary if the underlying cause of the opacity is contagious, but many causes do not require isolation (CDC, 2022).

- **Testing = 10**
  Comprehensive diagnostic evaluation is crucial for determining the exact cause of lung opacities, guiding targeted treatment and preventing misdiagnosis (Rubin et al., 2020).

## Key Takeaways

1. **Benefit vs. Harm**: The utility scores balance the potential benefits of medical interventions against their associated risks and costs.

2. **Disease-Specific Standard of Care**: Treatments are aligned with established clinical guidelines specific to each condition.

3. **Avoidance of Unnecessary Interventions**: The matrix discourages overtreatment in healthy individuals to prevent adverse effects and resource wastage.

Overall, the utility matrix aligns with standard clinical guidelines by advocating for appropriate treatment of infections, isolation of contagious diseases, thorough diagnostic evaluations for ambiguous findings, and avoiding unnecessary interventions in healthy patients.

