# OpenReview forum: "Decision Theoretic Foundations for Conformal Prediction: Optimal Uncertainty Quantification for Risk-Averse Agents"
_ICML.cc/2025/Conference — ICML 2025 spotlightposter_

### Official Review · Reviewer_jVCC · 2025-03-12

**Overall Recommendation:** 4

**Summary:**

This paper considers the marginal version of the value at risk problem. This problem is called risk-averse decision policy optimization.

- The authors derive optimal policy for risk-averse decision makers given prediction sets, which takes a max-min form.

- They establish that prediction sets are a sufficient statistic for safe decision-making.

- They characterize the optimal prediction sets for risk-averse conformal prediction by formalizing a risk-averse conformal prediction optimization problem.

- They propose a risk-averse calibration algorithm with a distribution-free safety guarantee.

## update after rebuttal:

I have read the author response as well as the other reviewers and author response to them. The authors clarified my major concerns related to the positioning of the paper with respect to the related work, technical novelty, group-conditional setting, and dependence on sample size. I would like to thank the authors for their good work. The changes that are mentioned in the author rebuttal should be incorporated into the final version of the paper.

**Claims And Evidence:**

It is said that the results are extendable to group conditional validity constraints. Perhaps this is the case with more realistic applications. It is said that the marginal formulation in the paper “naturally extends” to this case, but no roadmap is provided. It sounds as if this extension is trivial. If not, please specify what challenges lie ahead.

**Essential References Not Discussed:**

N/A.

**Experimental Designs Or Analyses:**

Experiments look valid.

**Methods And Evaluation Criteria:**

Yes.

**Other Comments Or Suggestions:**

N/A.

**Other Strengths And Weaknesses:**

I am positive about the contribution of this paper. However, a more detailed discussion is required on why conformal prediction sets are the right way to approach the problem (perhaps by comparing them with other approaches, like GP-based approaches in an experimental setting). In addition, novel steps of the technical proofs should be highlighted.

**Questions For Authors:**

1) This paper proposes a new model for risk-averse decision-making using conformal prediction sets. To avoid problems commonly encountered in conformal prediction, it considers the marginal version of the problem. In order to assess the technical contribution of the current work, please discuss if any new or out-of-the-box technique is used to arrive at the conclusions of the paper. Is it the reparameterization used to derive an equivalent reformulation of RA-CPO? Is it the structure discovered using duality?

2) Can any further insights be gained for the finite-sample setting? It seems that everything works fine when replacing true functions with their estimated counterparts. What is the effect of the sample size n on the quality of the estimates and prediction sets? Is it possible to get some convergence rates based on n?

**Relation To Broader Scientific Literature:**

Risk aversion and safety have been frequently studied by utilizing Gaussian processes as surrogate models. See, e.g.,

- Sui, Yanan, et al. "Safe exploration for optimization with Gaussian processes." International conference on machine learning. PMLR, 2015.

- Nguyen, Quoc Phong, et al. "Value-at-risk optimization with Gaussian processes." International Conference on Machine Learning. PMLR, 2021.

- Demirel, Ilker, et al. "Escada: Efficient safety and context aware dose allocation for precision medicine." Advances in Neural Information Processing Systems35, 2022.

Please comment on the pros and cons of your approach in risk-averse decision-making compared to this line of literature.

**Theoretical Claims:**

The theoretical claims make intuitive sense. I did not check the proofs.

---

> ### Author Rebuttal · Authors · 2025-03-31
>
> We thank the reviewer for their valuable feedback and thoughtful questions, which will help us significantly improve the clarity and contribution of our manuscript.
>
> **Question 1:**
> Our contributions are twofold, as outlined below: (1) we introduce a novel question within the conformal prediction literature, and (2) we develop mathematical techniques to provide meaningful answers.
>
> **Regarding the novel question:** Despite extensive literature on conformal prediction (CP) as uncertainty quantification beneficial for downstream decisions, little was previously known about whether prediction sets are the ideal means to communicate uncertainty within decision-making pipelines and how decisions should optimally incorporate them. We hope our findings bring new insights to the community in this regard.
>
> **Regarding mathematical and technical contributions:** A key contribution is deriving an explicit solution to RA-CPO (Section 2.2), accomplished in Proposition 3.1 and Theorem 3.2. To highlight the challenges, note that RA-CPO is a non-convex optimization over the set function $C(.)$, lying outside conventional duality-based analysis. We first reparametrize RA-CPO equivalently (Eq. 12), maintaining non-convexity. Next, in the proof of Theorem 3.2, through another reparametrization to a mixed-integer program, followed by a convex relaxation over function spaces, we use a key technical lemma (Lemma B.1, Appendix) to show this relaxation exactly solves RA-CPO. This approach is novel, and we will emphasize these contributions further in the revision.
>
> Additionally, our manuscript includes: Theorem 2.3 (prediction sets as sufficient statistics), Proposition 2.2 (optimality of max-min decision rule given marginally valid sets), and Theorem 4.1 (finite-sample statistical validity of our algorithm).
>
> **Question 2:**
> This is an excellent point. Two distinct estimation steps exist in the finite-sample algorithm: calibration-data-based and model-output-based (softmax probabilities). The only parameter influenced by calibration data (thus sensitive to sample size $n$) is the scalar parameter $\beta$. Similar to standard CP, the sensitivity to $n$ is minimal, stabilizing once $n$ reaches a few thousand samples. Under mild smoothness conditions on the conditional distribution of $Y|X$, finite-sample upper bounds scale as $1 - \alpha + O(1/n)$ (Theorem 4.1), a standard assumption in CP literature.
>
> Remaining computations, notably quantiles (Eq. 15), rely solely on softmax probabilities and are independent of calibration data and $n$, preserving statistical guarantees. We will further clarify this point by expanding the paragraph after Corollary 4.2.
>
> **Regarding group-conditional validity claims:**
> Although briefly noted in Remark 2.1, our original discussion was limited by space constraints. To clarify: the theoretical results in Sections 2 and 3—such as the characterization of optimal decision rules and prediction sets—extend naturally to the group-conditional setting, resulting in an $m$-dimensional formulation (where $m$ is the number of groups). However, the finite-sample algorithm in Section 4 poses greater challenges, as it requires calibrating an $m$-dimensional vector, which introduces both computational and statistical complexity. In the revised manuscript:
>
> - We have expanded Remark 2.1 to explicitly distinguish between the components that extend systematically and those that present additional challenges.
>
> - We have enhanced the “Future Work” section to outline potential approaches for addressing the difficulties in the finite-sample setting and to motivate further investigation of this important extension.
>
> **Regrading bayesian methods:**
> We thank the reviewer for highlighting relevant Bayesian literature. We emphasize our approach is complementary—not competitive—with Bayesian methods:
>
> Main theoretical contributions up to Section 3 (two-thirds of the paper) remain neutral regarding Bayesian or frequentist approaches, focusing instead on general roles of prediction sets in risk-averse decision-making. The equivalence formulation (Theorem 2.3) and optimal prediction set characterization (Theorem 3.2) assume complete knowledge of underlying distributions, applicable equally to Bayesian posterior distributions (e.g., from Gaussian Processes).
>
> In scenarios where Bayesian models (e.g., Gaussian Processes) approximate the distributions well enough, one can directly use our theoretical results without employing the finite-sample calibration of Section 4. Alternatively, even when Bayesian assumptions' precision is uncertain, one can still start from Bayesian posteriors and further calibrate prediction sets using our approach, ensuring robust safety guarantees.
>
> We will explicitly clarify these points in the revised manuscript, along with a discussion about the existing bayesian approaches, including the ones suggested by the reviewer.

---

### Official Review · Reviewer_DRsr · 2025-03-13

**Overall Recommendation:** 4

**Summary:**

This paper studies the decision-theoretic foundations of conformal prediction sets. It shows that prediction sets characterize the optimal strategy of risk-averse decision-making agents, and then connects it to a specific form of conformal prediction sets. Besides population-level characterization, based on the optimal prediction sets for this specific risk-averse strategy, methods for building finite-sample valid conformal prediction sets are established. The efficacy of the proposed methods is demonstrated by diverse numerical experiments.

**Claims And Evidence:**

Yes.

**Essential References Not Discussed:**

No to my current knowledge. See "questions" part for some conceptually related works.

**Experimental Designs Or Analyses:**

Yes.

**Methods And Evaluation Criteria:**

Yes.

**Other Comments Or Suggestions:**

N/A

**Other Strengths And Weaknesses:**

This paper is sound in general, and the results are interesting. However, I am not fully convinced by the marginal objective which is fundamental to the results throughout. Please see the "questions" section.

**Questions For Authors:**

1. Is the RA-DPO formulation equivalent to the per-x optimization? If so, it would be nice to point that out. (My understanding is no?)
2. Does the fact that C(x) is a random (set-valued) function change the interpretation of the minimax optimal policy result? How should one interpret these results in the context of conformal prediction, given that conformal prediction set is a random mapping?
3. Why is the objective chosen as the expectation of the lower bound $\nu(X)$? It seems risk-averse agents may also consider a tail-probability related quantity in $\nu(X)$, or the infimum of $\nu(X)$ over $X$?
4. Related to Question 1, changing from per-x problem to the marginal formulation makes the objective and constraints less interpretable. For example, asking for a marginally valid lower bound $\nu(X)$ gives opportunities to sacrifice some "hard" instances $X$, which (considering the average of $\nu(X)$) leads to something weird (equation 12), where $t(X)$ can be unequally distributed across $X$. I'm not sure whether this is still fully meaningful from a decision making perspective. Would it be possible to derive a marginal problem which is equivalent to the per-x ones?
5. In Section 4, instead of using a density estimator (which can be difficult especially for continuous outcomes), can one just directly estimate the conditional quantile of the utility for any action $a$, and let the prediction sets prioritize high-utility regions?
6. I like the perspective of decision making with conformal prediction, which is a prevalent issue but has thus far mostly been implicit and heuristic. I wonder how this perspective is related to several recent works with related concepts, where people use selective inference ideas to "pick out" units to act upon with false discovery/coverage rate control (e.g., [1,2]), or build prediction sets after decision making processes that have picked out some interesting units (e.g., [3,4]).

[1] Jin, Ying, and Emmanuel J. Candès. "Selection by prediction with conformal p-values." Journal of Machine Learning Research 24.244 (2023): 1-41.
[2] Gazin, Ulysse, et al. "Selecting informative conformal prediction sets with false coverage rate control." arXiv preprint arXiv:2403.12295 (2024).
[3] Bao, Yajie, et al. "Selective conformal inference with false coverage-statement rate control." Biometrika 111.3 (2024): 727-742.
[4] Jin, Ying, and Zhimei Ren. "Confidence on the focal: Conformal prediction with selection-conditional coverage." arXiv preprint arXiv:2403.03868 (2024).

**Relation To Broader Scientific Literature:**

How to use conformal prediction for actionable, transparent and reliable decision making is an important yet less noticed problem in the literature. This paper lays out theoretical results that may help advance research in this direction.

**Theoretical Claims:**

Yes. The theoretical results appear correct to me.

---

> ### Author Rebuttal · Authors · 2025-03-30
>
> We sincerely thank the reviewer for their careful and detailed evaluation, their supportive stance toward our work, and their insightful and constructive questions, which allow us to clarify and deepen our results significantly.
>
> **Questions 1, 3, and 4:**
> Thank you for these great questions! As the reviewer rightly points out, the most practically meaningful objective is the per-instance optimization (Eq. 2), which ensures optimal value-at-risk guarantees conditional on each individual instance $x$. However, there are two main considerations justifying our chosen marginal approach (RA-DPO):
>
> Distribution-free, finite-sample methods generally cannot produce simultaneous per-$x$ guarantees without additional assumptions. Hence, some form of marginalization is inevitable. Our specific choice, $E[\nu(X)]$, is indeed a deliberate choice we made, as there is no canonical choice and we had to fix an objective. We also agree with your observation that marginal guarantees might indeed result in uneven coverage across individual instances. To address precisely this issue, we included Remark 2.1, pointing toward extensions involving group-conditional guarantees, where validity can be enforced within predefined groups of covariates (e.g., patient demographics). Such group-based guarantees enable practitioners to ensure that important covariate subgroups retain meaningful coverage\risk. In the revised version, we will:
>
> 1) Extend Remark 2.1 to clearly show which parts of our theory can be systematically extended to group-conditional scenarios and which present challenges.
>
> 2) Expand our "Future Work" section to highlight that while Sections 2 and 3 extend naturally to $m$-groups setting (yielding an m-dimensional characterization), the finite-sample algorithm in Section 4 faces additional challenges due to the complexity of calibrating an m-dimensional vector (along with potential approaches to address these challenges).
>
> It is also worth mentioning that moving to group-conditional guarantees, also makes our framework less sensitive to the choice of $E[\nu(X)]$, as eventually in the fully conditional setup, the problem reduces to per-x problems.
>
> We will explicitly clarify these points, naming the deliberate choice of $E[\nu(X)]$ compared to alternatives and the important role of group conditional guarantees for a more meaningful decision making pipeline, in Section 2 of our revised version.
>
> **Question 2:**
> To clarify, we focus on deterministic prediction sets in this paper, meaning a deterministic map from X to sets. Also in CP, prediction sets are deterministic when calibration data is fixed (which is the case in practice). Although, due to their mathematical convenience, the CP guarantees are usually formulated as an expectation over calibration data (similar to our Theorem 4.1), but one can also derive PAC guarantees for the same algorithm, which provides coverage guarantees conditional on the calibration data.  We will discuss this important matter in the revised version.
>
> **Question 5:**
> Thank you for highlighting this important practical question. Indeed, in Section 4, our current algorithm assumes a predictive model providing softmax probabilities, implicitly targeting classification tasks. Several promising approaches can be adopted in practice:
>
> As you suggest, one natural approach would involve training separate quantile-prediction models for each action and then choosing among them. Alternatively, a single regression model predicting the maximum quantile over all actions (as defined in Eq. 9) could also be learned directly. Both of these are similar ideas to Conformalized Quantile Regression (CQR). Other methods, such as regression-diffusion models that generate multiple samples per instance, could also approximate the quantiles in Eq. 9. We will leverage the additional two pages allowed to discuss explicitly these practical regression approaches and provide guidelines for practitioners.
>
> Thanks again for providing an interesting approach for handling regression.
>
> **Question 6:**
> We are grateful for bringing up this valuable literature. We see selection conditional conformal prediction as a complementary line of works to ours.
>
> In the revised version, we will include a paragraph discussing this direction including the works you pointed out ([1-4]). Briefly, selective inference addresses scenarios where multiple prediction sets are constructed, and one selects a subset of these sets (e.g. choosing proteins with high predicted affinity), which then creates selection bias that one has to account for. In contrast, our current formulation constructs a single prediction set per instance $x$, with the action directly derived via a max-min strategy from that set, thereby avoiding selection bias.
>
> An interesting future work could be to look at the intersection of these works where one might want to select a subset of actions, while both remaining statistically valid and optimizing for a form of value at risk.

---

### Official Review · Reviewer_q2pE · 2025-03-14

**Overall Recommendation:** 4

**Summary:**

This paper aims to address three questions at the intersection of conformal prediction (CP) and decision making: (1) Understanding what type of uncertainty quantification is best for risk-averse decision makers, (2) How risk-averse decision makers should use prediction sets (ie, what policy they should use, given a prediction set with marginal coverage validity), and (3) How to design CP sets for such risk-averse decision makers. In this paper, “risk-averse decision making” refers to selecting an action $a$ in response to context $x$ that maximizes the “value at risk” $\nu_{\alpha}(a; x)$: that is, maximizing the smallest value/utility that the agent can expect to receive with high probability $1-\alpha$ (ie, the probability/risk of receiving utility less than $\nu_{\alpha}(a; x)$ is at most $\alpha$).

To address questions (1) and (2), the authors provide analysis and discussion that connects the objective of (marginal) risk-averse decision making (i.e., Risk Averse Decision Policy Optimization or “RA-DPO”) with one of risk-averse decision-making using conformal prediction sets (i.e., Risk Averse Conformal Prediction Optimization or “RA-CPO”): That is, the authors prove that from any optimal solution of RA-DPO one can construct an optimal solution to RA-CPO with the same utility, and vice versa (Theorem 2.3). To address question (3), the authors provide some analysis for the optimal prediction sets (Section 3) and a practical algorithm building on this analysis (Section 4), and they evaluate the proposed algorithm compared to an expectation-maximizing or “best-response” policy baseline on medical diagnosis and recommender system tasks (Section 5).

## update after rebuttal (Note: I copy-pasted this from my response to the authors' rebuttal.)

Thank you for your detailed response to my questions and concerns! In particular: The concrete changes that the authors describe address the main concern I had around the framing and references; the expanded remark and discussion they describe on the distinction between marginal and conditional seems like it will be valuable for future readers (and avoiding common points of confusion on this distinction); and the explanation regarding why selection bias is avoided (ie, the quantiles are taken for each action [based on softmax outputs from the model], independently of the calibration data) was helpful to me.

Re notation clarity, I also appreciate the clarification about why $\nu$ and $\nu_\alpha$ are distinct (ie, the first being the optimization variable), and the different roles of $t$ and $alpha$. I still feel like $\theta$ was perhaps unnecessary notation, and maybe something clearer could be achieved with something like $\nu_t$ or $\nu_{1-t}$ (ie, only changing the subscript variable), but this is a stylistic comment and ultimately up to the authors. A proof-read for a camera-ready version could help double-check that all notation and new terms (eg, $\nu$ vs $\nu_\alpha$) are clearly introduced with their distinctions, and generally to address other points of confusion/questions among the reviews.

I'm happy to update my score--congrats on a nice paper!

**Claims And Evidence:**

Mostly yes: All of the main formal theoretical claims appear to be solid, the experimental evaluations appear to be reasonable and sound, and the high-level takeaway from the paper should be very valuable to the broader CP and ML communities (i.e., roughly that “prediction sets are a good form of uncertainty quantification for risk-averse decision making”).

However, I have a couple concerns or questions about main claims:
- **(1) Ensuring proper contextualization relative to related work:** For example, although there are many conformal prediction papers cited in the Related Work (Sec 1.1) and Further Related Work (Appendix A), I do not see any references to or discussion of classic literature on risk-averse statistical decision making, which there is much of, e.g., in economics (see “Relation To Broader Scientific Literature” and “Essential References Not Discussed”). Additionally, the Further Related Work may miss some CP references more closely related to decision making while instead citing many CP papers that are influential but not as relevant to the current paper.
- **(2) “Minding the gap” between marginal and conditional objectives/decision making:** The paper does acknowledge the gap between the *conditional* objectives/decision making/coverage guarantees that one would ideally like and the “Marginal Version” (see Sec 2) that is realized in practice with CP, but there are some parts of the methods section where this distinction seems less clear, and the paper seems to lack a thorough discussion of the limitations of marginal decision-making, which would be valuable for readers, practitioners, and future researchers building on this work.
- **(3) Question about whether using same calibration set to select action affects CP guarantee:** See “questions for authors”

**Essential References Not Discussed:**

**Literature on risk-averse decision making:** Apart from citing Duffie and Pan 1997 to introduce the definition of “Value at Risk” in Section 2, there appears to be no mention or discussion of prior literature on risk-averse decision making (eg, https://en.wikipedia.org/wiki/Risk_aversion); I think it is essential to at least mention/acknowledge this literature, if not discuss some connections more thoroughly in the appendix. Eg, whereas currently Sec 2 “Fundamentals of Risk Averse Decision Making” could be read by many reasonable readers as if this paper is inventing this topic, I think there should be some mention of prior literature such as the following, and potentially further discussion in appendix. Following are some examples:
- *Arrow-Pratt measure of absolute risk aversion / coefficient of absolute risk aversion:*
- Arrow, K.J. (1965) Aspects of the Theory of Risk Bearing
- Pratt, J. W. (1978). Risk aversion in the small and in the large. In Uncertainty in economics (pp. 59-79). Academic Press.

- *Fundamental connections between second-order stochastic dominance and risk aversion (e.g., see https://en.wikipedia.org/wiki/Stochastic_dominance#Second-order):*
- Hadar, J., & Russell, W. R. (1969). Rules for ordering uncertain prospects. The American economic review, 59(1), 25-34.
- Meyer, J. (1977). Second degree stochastic dominance with respect to a function. International Economic Review, 477-487.

**Arguably highly relevant CP+decision making papers not cited/discussed:**
- *Papers on CP for counterfactual treatment-effect estimation for personalized decision making [paper-specific context in brakcets]:*
- [CP for counterfactual treatment assignment decisions] Lei, L., & Candès, E. J. (2021). Conformal inference of counterfactuals and individual treatment effects. Journal of the Royal Statistical Society Series B: Statistical Methodology, 83(5), 911-938.
- [Extension for robustness to potential confounding] Yin, M., Shi, C., Wang, Y., & Blei, D. M. (2024). Conformal sensitivity analysis for individual treatment effects. Journal of the American Statistical Association, 119(545), 122-135.
- [Extension for robustness to potential confounding] Jin, Y., Ren, Z., & Candès, E. J. (2023). Sensitivity analysis of individual treatment effects: A robust conformal inference approach. Proceedings of the National Academy of Sciences, 120(6), e2214889120.

- *Papers on CP under distribution shifts induced by decision making/actions of an AI/ML agent (e.g., black-box optimization or contextual bandits):*
- [CP for contextual bandits decision making, off-policy setting] Taufiq, M. F., Ton, J. F., Cornish, R., Teh, Y. W., & Doucet, A. (2022). Conformal off-policy prediction in contextual bandits. Advances in Neural Information Processing Systems, 35, 31512-31524.
- [Theory & expts on CP under feedloop shifts induced by active learning or black-box optimization ML decision process] Prinster, D., Stanton, S., Liu, A., & Saria, S. (2024, July). Conformal validity guarantees exist for any data distribution (and how to find them). In Proceedings of the 41st International Conference on Machine Learning (pp. 41086-41118).
- [Empirically studies CP under feedback-loop shifts induced by BayesOpt decision making] Stanton, S., Maddox, W., & Wilson, A. G. (2023, April). Bayesian optimization with conformal prediction sets. In International Conference on Artificial Intelligence and Statistics (pp. 959-986). PMLR.
- [CP under one-step “feedback covariate shift” decision making] Fannjiang, C., Bates, S., Angelopoulos, A. N., Listgarten, J., & Jordan, M. I. (2022). Conformal prediction under feedback covariate shift for biomolecular design. Proceedings of the National Academy of Sciences, 119(43), e2204569119.

It feels important to mention many if not all of these, especially given that in Appendix A (second to last paragraph) there currently appears to be superfluous (ie, 5 whole lines of in-text citation) references to many CP papers that are (in my view) far less related to decision making than the ones I have provided here....

**Experimental Designs Or Analyses:**

I overall reviewed the experimental settings and they seem to be reasonable. E.g., it was important that the experiments not only be regression or classification, but also have some sort of actions and corresponding utilities (e.g., Table 1 for medical diagnosis experiment).

**Methods And Evaluation Criteria:**

Overall, the experimental settings and evaluations seem to sufficiently support the main message of the paper, that is regarding how prediction sets and the proposed Risk-Averse Calibration (RAC) algorithm can be used for risk-averse decision making.

That said, it appears that the only baseline is an expectation-maximizing “best response” policy, and it would likely improve the paper if the proposed RAC algorithm were compared with other methods that use CP for decision-making--such as methods proposed in Vovk and Bendtsen (2018), who have a method based on conformal predictive distributions for asymmetric classification, similar to the experimental setting in this paper, or perhaps methods in Lekeufack et al. (2024), if relevant--however, since I view the main message of the paper to be one that’s more broad and conceptual regarding prediction sets, I don’t view this as absolutely necessary.

**Other Comments Or Suggestions:**

- Subheadings could sometimes be added or revised for clarity. Eg., prior to reading the sections closely, it initially thought sections 2.1 (“A Prediction Set Perspective”) and 2.2 (“An Equivalent Formulation via Prediction Sets”) seemed very similar.
- In Section 3, the paragraph before Prop 3.1, should $Pr(Y \in C(x))\geq t$ actually be $Pr(Y \in C(x) | X = x)\geq t$? As, the latter is what is written in Prop 3.1, and in the paragraph it says the probability is over $p(y|x)$.
- In Section 3, I found it confusing whether $t$ needed to be referring to conditional coverage, or whether it could be marginal coverage. That is, $t$ is introduced as “conditional coverage probability,” but then in (12) it is used as a marginal coverage constraint.

**Other Strengths And Weaknesses:**

**Other Strengths:** Overall, the paper provides an insightful perspective on the connections between prediction sets and risk-averse decision making. Additionally, the writing style is very appealing for a wide audience, with broader implications that can be understood beyond the scope of conformal prediction, so it seems like it has potential for high impact.

**Other Weaknesses / opportunities for improvement:**
- *Occasionally oversimplified notation: Eg, $\nu(X)$ should ideally be $\nu_{\alpha}(X)$ in many instances including RA-DPO def:* The paper seemingly makes an effort to simplify notation for readability, but at times this may be oversimplified, which can make close-reading of technical details difficult, confusing, or perhaps misleading. For example, the value-at-risk quantity $\nu_{\alpha}(a; x)$ is defined to depend on $\alpha$, $a$, and $x$, but sometimes it is stated only as $\nu$, sometimes it is stated as $\nu_{\alpha}(x)$, and sometimes it is stated as $\nu(x)$. Sometimes this is done explicitly (eg, Eq (2)), but other times, eg, the $\alpha$ is dropped casually without explanation. In my view, this is not without consequence, because I think that $\nu(x)$ actually depends on $\alpha$ as $\nu_{\alpha}(x)$ makes Theorem 2.3 far less “surprising.” That is, just prior to Theorem 2.3, the paper states “One might expect a-priori that passing from the actual distribution to a lossy prediction set representation would discard information that is critical to finding the optimal policy. However, the following theorem shows, perhaps surprisingly, that this is not the case…”; however, once one realizes that in the RA-DPO def that $\nu(X)=\nu_{\alpha}(X)$, it is easier to see how RA-DPO is effectively only leveraging the level-$\alpha$ quantile of the full utility distribution, i.e., the same information provided by the prediction set.
- Limitations of focusing on marginal vs conditional objective could be discussed further.
- *Potentially redundant notation (?):* It is unclear to me why $\theta(x, t) := max_{a\in A}\ quantile_{1-t}[u(a, Y) | X=x]$ (Eq. (9)) needed to be introduced as a new quantity that is distinct from $\nu_{\alpha}(x) := max_{a\in A}\ quantile_{\alpha}[u(a, Y) | X=x]$ (Eq. (2)), as clearly they are equivalent when $1-t=\alpha$. An explanation is welcome, but so far my impression is that this difference in notation has only caused confusion for me, and perhaps served to make the reformulation in Eq. (12) seem more “surprising”...
- *Potentially missed opportunity to draw connections with choice of conformity score(?):* Where the prediction set $C^*(x)$ is defined in Eq. (13) and similarly for $\hat{C}(x;\beta)$ at the end of page 6, to a CP-minded reader, it seems like the optimal/proposed prediction sets are defined in a way that would be equivalent to having conformity scores corresponding to the utility function, for an optimal/chosen action (or from a pessimistic viewpoint, could be nonconformity score corresponding to a cost/loss function). In other words, it seems like a takeaway from the paper could be rephrased for a CP-minded person as ‘design your (non)conformity score to correspond to utility (or loss) that you care about for decision making, and then pick action/policy s.t. quantile on the utility (or loss) is best.’

**Questions For Authors:**

**Question about whether there is selection bias that may affect CP guarantee (?)** At a high level, eg based on the max-min decision rule, Figure 1, and the prediction sets defined in Eq (11) and (13), it seems like the big picture story from a CP view is: (i) Chose conformity scores that correspond to utilities, (ii) for each action, compute utility distribution over calibration data and take an appropriate quantile to get the value-at-risk for that action, and (iii) select the action with maximum estimated value-at-risk for the worst-case label. Is this roughly accurate, or is there a more accurate way to rephrase the proposed procedure in the terminology connected to conformity scores? In either case, I am wondering if there is an issue of (or if so, how it is avoided) of creating selection bias, that is when an action is taken based on observing score distribution over the same holdout calibration set, I think this could invalidate CP guarantees, as in the potentially analogous case of selecting a model based on efficiency of CP sets, eg, see: Liang, R., Zhu, W., & Barber, R. F. (2024). Conformal prediction after efficiency-oriented model selection. arXiv preprint arXiv:2408.07066. I appreciate and look forward to your response!

**Rationale for current recommendation:** Broadly, I view the paper as having an insightful contribution connecting prediction sets to risk-averse decision making. I have attempted to do a detailed review of the paper for correctness, clarity, and relation to broader literature, among other factors, in part due to its potential. I want to recommend accepting this paper due to the valuable perspective it offers and broad potential appeal, but at the moment I think it my concerns make me think it is most prudent not to at least until I have heard more from the authors at the discussion period. To reiterate my concerns, the main one is about further stating relation to literature (especially prior work on risk-averse decision making, and secondarily about CP+decision making papers that may be more relevant than many CP refs provided in App A: Related Work); other concerns are those stated around some notational clarity and the above question on possible selection bias. If these concerns are addressed, I’d be happy to consider improving my score.

**Relation To Broader Scientific Literature:**

There is growing work on leveraging conformal prediction (CP) to improve decision making, and understanding the theory of doing so; separately, there is also a large literature on risk-averse statistical decision making, though admittedly much of it is in the economics, finance, and game theory literature (eg: https://en.wikipedia.org/wiki/Risk_aversion). In my view, this paper draws insightful and useful connections between CP/prediction sets and risk-averse statistical decision making, although it unfortunately seems to not discuss this latter literature at all (aside from citing Duffie and Pan 1997 to introduce the definition of “Value at Risk”); and regarding the CP literature, there are arguably highly relevant references missing, and superfluous references provided. Please see the following “Essential References Not Discussed” Section for specific details.

**Theoretical Claims:**

Yes, I made an effort to check the proofs and they appear to be sound, though it is possible there are some details I did not understand given time constraints.

---

> ### Author Rebuttal · Authors · 2025-03-26
>
> We sincerely thank the reviewer for their detailed and insightful feedback. We are grateful for the positive recognition of our core contributions—especially the novel linkage between prediction sets and risk-averse decision-making. In what follows, we detail the concrete actions we have taken to address the reviewer’s concerns.
>
> **References:**
> The reviewer’s comment on expanding the references is extremely valuable. We agree that including prior work from the economics literature on risk aversion is crucial to understanding the challenges and collective effort in risk-sensitive decision making. In the revised version:
>
> 1) We have expanded the Related Work section by adding two new paragraphs. One discusses the economics and game theory literature (including all the suggested references) to highlight classical approaches to risk aversion. The other covers additional works, including the ones mentioned by the reviewer, in CP that address decision making beyond risk control.
>
> 2) We fully acknowledge that we are by no means claiming to have invented the topic of risk-averse decision-making, but rather are drawing on concepts developed in the economics literature. To better reflect this, we have renamed the section “Fundamentals of Risk-Averse Decision Making” to “Preliminaries of Risk-Averse Decision Making.” In this section, we now include an additional paragraph—along with appropriate citations to the economics literature—that briefly outlines alternative formulations of risk aversion.
>
> We again thank the reviewer for highlighting relevant work from the literature, which has been very helpful in improving our presentation. While space constraints prevent us from discussing detailed connections to these works, we would be happy to explore them further in follow-up rounds.
>
> **Marginal and Conditional Objectives:**
> We thank the reviewer for emphasizing this important issue. Although Remark 2.1 highlighted this gap, space constraints limited our discussion in the original submission. In finite sample settings it is impossible to learn an optimal action for every individual scenario without some type of averaging over the covariate space. A common approach in CP is to use group-conditional guarantees, where validity is ensured on average over pre-specified groups (e.g., based on patient age or sex). We have:
>
> 1) Extended Remark 2.1 to clearly delineate which parts of our theory can be systematically extended to group-conditional scenarios and which present challenges.
>
> 2) Expanded the Future Work section to discuss these issues in detail. Specifically, we explain that while the results in Sections 2 and 3 extend naturally to the group-conditional setting (resulting in an m-dimensional characterization), the finite sample algorithm in Section 4 becomes more challenging due to the need to calibrate an m-dimensional vector.
>
> **Conformity Scores and Selection Bias:**
> We appreciate the reviewer’s thoughtful questions on these topics. Regarding the connection to conformity scores, it is crucial to note that our optimal prediction sets are not derived from calibrating a threshold for a conformity score. As shown in Section 3, the optimal sets do not form a nested sequence as the miscoverage threshold varies. In contrast, optimal sets for objectives like minimizing average set size are typically characterized by a threshold rule (e.g., $p(y∣x)≥q$). We will add a remark to clarify this technical difference, which is crucial for deriving optimal actions for risk-averse agents.
>
> We do not suffer from selection bias. We construct one prediction set per test instance and apply the max-min rule directly—rather than selecting among multiple candidate sets. The calibration procedure involves a one-dimensional parameter, $\beta$, which is determined using calibration data to ensure valid coverage (see Theorem 4.1). Although our calibration phase involves computing a quantile for each action (as in eq. (15)), these computations rely on the softmax outputs from the model and remain independent of the calibration data, preserving the statistical guarantees.
>
> **Notational Clarity:**
> Note that $\nu$ and $\nu_\alpha$ are distinct: $\nu_\alpha$ is defined in Eq.(2) while $\nu$ is the optimization variable in RA-DPO. In RA-DPO, the decision maker jointly optimizes for a utility certificate ($\nu$) and an action policy ($a$). This distinction clarifies why Theorem 2.3 is non-trivial, as it is not obvious that the optimal utility certificate can be obtained as the max-min value of a prediction set.
>
> Regarding the notations $\theta$ and $\nu_\alpha$, while they are equivalent when $1-t = \alpha$, it is important to note that $t$ and $\alpha$ play different roles. Here, $\alpha$ is the fixed miscoverage threshold (as in standard CP), whereas $t$ is a tunable variable during calibration to ensure that the final prediction sets achieve the desired coverage. We will clarify this point in the revised version to prevent confusion.

---

### Official Review · Reviewer_c5sw · 2025-03-14

**Overall Recommendation:** 3

**Summary:**

The paper studies the decision-theoretic properties of conformal prediction sets. For risk-averse agents who want probabilistic certificates on certain actions for the utility to be greater than some value with high-probability, the paper suggests that this goal can be accomplished via conformal prediction sets that satisfy marginal coverage guarantees, in fact they show that both notions can be derived from each other, establishing that prediction sets are fundamental objects for a risk-averse agent. The paper then proposes an algorithm to devise prediction sets that result in better utility for risk-averse agents, with experiments supporting the claims.

**Claims And Evidence:**

The paper provides enough support for the claims made. However, I'd appreciate more justification for the max-min policy for the agent (Equation 7). While it makes sense, and frankly an intuitive thing to do, but I assume a decision-maker might be interested in resolving the uncertainty over $\Omega$ in a different way (e.g. from here https://arxiv.org/abs/2204.11318). So it makes sense to justify the max-min rule as employed, as why that is used. I'll frame my question differently: Instead of "we are interested in the policy that is minimax-optimal" (line 191), I appreciate the answer to "why we are interested in the policy that is minimax-optimal?"

**Essential References Not Discussed:**

None

**Experimental Designs Or Analyses:**

The experiments sound convincing, and make sense. However, I'd appreciate some reasoning why RAC better aligns nominal miscoverage with realised miscoverage  (Figure 2d), as compared to other scores.

**Methods And Evaluation Criteria:**

Yes, the paper tests their methodology appropriately.

**Other Comments Or Suggestions:**

None

**Other Strengths And Weaknesses:**

The paper could be heavily improved in terms of writing, in particular Section 3 and Section 4. More intuition can be provided behind Algorithm 1. I'd appreciate how Algorithm 1 gets rid of specifying utility as everything before that $\hat{\theta}$ and $\hat{a}$ are defined in terms of the underlying utility of the decision-maker agent?

**Questions For Authors:**

I get the approach requires specifying the miscoverage level $\alpha$. Is there a way for it to be connected to the level of risk aversion of a decision-maker, or how should the decision-maker choose $\alpha$ depending on their level of risk-aversion. I see that RAC starts matching the averaged realised utility for nominal miscoverage level around 0.1 (Figure 2c), which makes sense that the prediction set sizes  might gets shrink and in the extreme case, only the true label will be in the prediction set, in which case the expected utility and the utility realised by RAC will coincide? But it does not serve the risk averse individual as they are not concerned with the average utility? I'd appreciate any clarification on this; it could be possible that my question is poorly worded. I guess I'm interested in how to design the sets that balances the risk aversion of a decision-maker with their utility maximisation?

**Relation To Broader Scientific Literature:**

The paper is certainly relevant to the ICML community. The paper advances the decision-theoretic interpretation of conformal sets, as in how to use them, and devises an algorithm to use them better in that sense.

**Theoretical Claims:**

I have verified all the statements to the point that I agree on a high-level that they are correct, but haven't fully verified the notational writing. However, I haven't formally checked the duality arguments (or the validity of them) that leads to Equation 14.

---

> ### Author Rebuttal · Authors · 2025-03-30
>
> We sincerely thank the reviewer for their insightful comments, constructive feedback, and positive evaluations of our paper.
>
> **On the motivation for minimax optimality:**
> This is an important point, and we appreciate the opportunity to clarify our choice. As you correctly pointed out, minimax optimality is indeed not the only conceivable approach. However, our rationale for selecting it is both intuitive and theory-backed, which can be summarized as follows:
>
> 1) From an intuitive standpoint, minimax optimality aligns well with the perspective of a risk-averse decision maker. Specifically, the set $\Omega$ captures all plausible scenarios faced by the decision maker, and a risk-averse agent would reasonably seek actions that are robustly optimal in the worst-case realization among these scenarios. This conservative stance naturally leads to the adoption of a minimax criterion.
>
> 2) Perhaps a more complete picture emerges by looking at Theorem 2.3. There we prove the max-min rule (introduced in Section 2.1), combined with optimized prediction sets, matches the optimal solution to the risk-averse decision problem (RA-DPO). In light of that, the minimax optimality of the max-min rule (proved in Proposition 2.2) states that for any fixed marginally valid prediction set, even if the prediction sets are not optimally derived, the max-min rule is still a natural choice. This is particularly important as in practice, due to the limitations of finite sample data, we can only have "approximately" optimal prediction sets.
>
> Nevertheless, we fully acknowledge your remark that alternative decision-theoretic approaches (such as those discussed in the paper you mentioned, which we will cite in the revised version) are also valuable and potentially yield interesting complementary results. We will incorporate a remark explicitly addressing this.
>
> **On the comment regarding improvements in presentation:**
> We completely agree that the clarity and intuitive presentation of our results could significantly benefit from additional explanations. Indeed, this has largely been limited by the strict page constraints. To address this, we will leverage the two extra pages allowed for the revised version to provide detailed intuition and clearer explanations, especially regarding:
>
> 1) The duality derivations and additional implications arising in Section 3.
>
> 2) Intuitive justification and step-by-step reasoning behind Algorithm 1 in Section 4.
>
> Regarding Algorithm 1's connection to utility, the utility function enters through the definition of $\hat{C}$ just before Algorithm 1. To summarize the intuition briefly here: Algorithm 1 uses the predictive model to approximate key quantities required for deriving optimal prediction sets, up to a scalar calibration parameter obtained from calibration data. As highlighted in the paragraph following Corollary 4.2, the safety guarantees of actions produced by our framework hold regardless of predictive model quality. Nonetheless, higher-quality models produce actions closer to the optimal choices among the safe ones.
>
> **On RAC’s coverage alignment:**
> We believe this phenomenon arises due to finite-sample effects. For instance, in the same Figure 2(d) on the left, one can see that RAC's miscoverage is sometimes better and sometimes worse. Moreover, it is important to emphasize that RAC differs algorithmically from conventional CP methods, which rely primarily on thresholding conformity scores. Consequently, finite-sample behaviors may differ across these approaches. However, as the data size grows sufficiently large, we anticipate these differences will diminish, and coverage behaviors will become comparable across all CP methods, given their common coverage guarantees.
>
> **On tuning the miscoverage level $\alpha$:**
> Thank you for this insightful question—this is indeed a critical practical concern. In our framework, the value of $\alpha$ is treated as a given, provided alongside the utility function. Accordingly, RAC takes $\alpha$ as an **input**. As you pointed out, this value should be selected by the decision maker to reflect their degree of risk aversion, and choosing an appropriate $\alpha$—much like specifying a utility function—can be non-trivial in some applications.
>
> From a practical standpoint, one can evaluate several candidate values of $\alpha$ using validation data, and the decision maker can select the one that offers the best trade-off between risk aversion and utility. This process can involve inspecting plots similar to those presented in our paper. For example, in the context of medical treatment recommendations, one could tune $\alpha$ by jointly examining Figures 2(b) and 2(c), and choosing a value that strikes the desired balance between maximizing average utility and minimizing critically harmful treatment decisions.
>
> We will add a remark in the revised version to clarify this point.

---

> > ### Comment · Reviewer_c5sw · 2025-04-05
> >
> > thanks for the detailed response. I agree with it. However, one concern is still not answered. In Figure 2c, as \alpha is increased, RAC approaches the best respond policy in terms of the average realised utility? What is the rationale for that? Could that be clarified?

---

> > > ### Author Response · Authors · 2025-04-07
> > >
> > > We thank the reviewer for their thoughtful engagement with our work. We appreciate the opportunity to clarify and improve the presentation of our paper.
> > >
> > > Regarding Figure 2c and the observation that RAC approaches the performance of the best-response method in terms of average utility as $\alpha$ increases, the intuition is as follows:
> > >
> > > The parameter $\alpha$ controls the level of risk aversion in our RAC method. With smaller values of $\alpha$, RAC prioritizes avoiding critical errors (situations resulting in extremely low utility) by opting for more conservative actions, typically leading to medium-level utility outcomes. This conservative approach naturally sacrifices the possibility of achieving higher but riskier utilities.
> > >
> > > However, as we increase the value of $\alpha$, RAC progressively allows for riskier actions by shrinking the size of the prediction sets. Eventually, when these prediction sets predominantly contain only a single item—corresponding to the highest likelihood prediction from the model—the decision-making process of RAC essentially reduces to the best-response policy. In this scenario, the max-min decision rule simplifies directly to choosing the best response to that single prediction.
> > >
> > > To further illustrate this behavior, we will include the following plot in the revised version of our paper: https://ibb.co/WCRbxwK
> > >
> > > Specifically, we will provide histograms depicting the full distribution of realized utilities at test time for three variants of RAC (with varying $\alpha$ values) compared directly against the best-response method. These histograms clearly demonstrate how adjusting $\alpha$ effectively controls the tails of the utility distribution and how, with increasing $\alpha$, RAC's realized utilities initially mirror those obtained by the best-response approach but eventually decrease if $\alpha$ is chosen excessively large.
> > >
> > > It is also important to note that excessively increasing $\alpha$ after some point can actually degrade the average utility. This happens because RAC always ensures the marginal coverage of the sets matches $1 - \alpha$. Therefore, if $\alpha$ becomes too large such that $1 - \alpha$ is less than the model's accuracy, RAC is forced not to choose the single prediction with the highest likelihood—since doing so would yield a marginal coverage greater than $1 - \alpha$. Instead, RAC might select the next best likelihood or produces empty prediction sets, both of which negatively impact average utility. That is to say, tuning $1-\alpha$ below the test accuracy of the model might not be a good idea.
> > >
> > > We believe the addition of this plot can enhance the presentation of our paper. Thank you again for your helpful comments and support.

---

### Decision · Program_Chairs · 2025-05-01

**Decision:**

Accept (spotlight poster)

**Comment:**

This paper presents a decision-theoretic framework for risk-averse prediction using prediction sets. It defines optimal uncertainty quantification, proposes a max-min policy for action selection, and introduces Risk-Averse Calibration (RAC), a practical algorithm that balances safety and utility. Experiments show RAC’s effectiveness in medical diagnosis and recommendation systems. All reviewers recommended acceptance, recognizing the paper’s strong theoretical contributions and practical relevance. The minor concerns raised were satisfactorily addressed in the rebuttal. Overall, this is a well-executed and timely contribution, and I strongly support its acceptance.